# Suppression weakens unwanted memories via a sustained reduction of neural reactivation

Ann-Kristin Meyer*, Roland G Benoit*

Max Planck Institute for Human Cognitive and Brain Sciences, Leipzig, Germany

**Abstract** Aversive events sometimes turn into intrusive memories. However, prior evidence indicates that such memories can be controlled via a mechanism of retrieval suppression. Here, we test the hypothesis that suppression exerts a sustained influence on memories by deteriorating their neural representations. This deterioration, in turn, would hinder their subsequent reactivation and thus impoverish the vividness with which they can be recalled. In an fMRI study, participants repeatedly suppressed memories of aversive scenes. As predicted, this process rendered the memories less vivid. Using a pattern classifier, we observed that suppression diminished the neural reactivation of scene information both globally across the brain and locally in the parahippocampal cortices. Moreover, the decline in vividness was associated with reduced reinstatement of unique memory representations in right parahippocampal cortex. These results support the hypothesis that suppression weakens memories by causing a sustained reduction in the potential to reactivate their neural representations.

## Editor's evaluation

Meyer and Benoit explore the brain mechanisms underpinning memory suppression. The authors demonstrate that suppressed memories undergo sustained fading over time, which diminishes the vividness of the memory representation, and is associated with reduced reactivation of the memory representations in the parahippocampal cortex.

*For correspondence:
annmeyer@cbs.mpg.de (A-KM);
rbenoit@cbs.mpg.de (RGB)

**Competing interest:** The authors declare that no competing interests exist.

## Introduction

Memories of the past are not always welcome. There are experiences that we would rather not think about, yet that involuntarily intrude into our awareness. Research over the last two decades has demonstrated that we are not at the mercy of such unwanted memories: we can control them by actively suppressing their retrieval (*Anderson and Hulbert, 2021*; *Fawcett and Hulbert, 2020*; *Norby et al., 2010*). This process weakens the memory and can eventually cause forgetting (*Stramaccia et al., 2021*). Here, we seek to tie the sustained phenomenological weakening of a suppressed memory to its neural basis.

Neuroimaging research has made strides in determining the transient neural mechanisms that prevent unwanted retrieval. It has consistently shown that retrieval suppression is associated with increased activity in right dorsolateral prefrontal cortex (dlPFC) and decreased activity in the hippocampus (*Anderson and Hanslmayr, 2014*; *Benoit et al., 2015*; *Benoit et al., 2016*; *Benoit and Anderson, 2012*; *Depue et al., 2007*; *Gagnepain et al., 2014*; *Gagnepain et al., 2017*; *Mary et al., 2020*; *Paz-Alonso et al., 2013*). This pattern has been interpreted as a top-down inhibition of critical hippocampal retrieval processes by the dlPFC (*Anderson and Hanslmayr, 2014*; *Benoit and Anderson, 2012*; *Depue et al., 2007*).

During retrieval, the hippocampus is integral for reinstating the cortical activity patterns that were present during the encoding of the memory (*Liu et al., 2012*; *Neunuebel and Knierim, 2014*; *Ritchey et al., 2013*; *Wing et al., 2015*; *Xue et al., 2010*). Inhibition of the hippocampus would accordingly hinder such momentary cortical reactivation and thus prevent unwanted retrieval (*Anderson and Hanslmayr, 2014*; *Gagnepain et al., 2014*). Consistent with this account, retrieval suppression has been found to also affect activity in cortical regions that encode the particular content of the suppressed memory (e.g. *Benoit et al., 2016*; *Depue et al., 2007*; *Gagnepain et al., 2014*; *Gagnepain et al., 2017*; *Mary et al., 2020*).

For example, when the unwanted memories comprise images of complex scenes, suppression is accompanied by a transient reduction of activation in the parahippocampal cortex (PhC) (*Benoit et al., 2015*).This region particularly supports memories for scenes (*Bohbot et al., 2006*; *Horner et al., 2015*; *Staresina et al., 2011*; *Staresina et al., 2013*) and its activity during retrieval scales with the detailedness (*Qin et al., 2011*; *Tendolkar et al., 2008*) and vividness (*Kensinger et al., 2011*; *Sheldon and Levine, 2013*; *Todd et al., 2013*) of the memories. Moreover, more fine-grained analyses of the activity patterns within the PhC have linked the reactivation of memory-specific representations to the successful retrieval of scenes (*Martin et al., 2013*; *Staresina et al., 2012*; *Staresina et al., 2013*). In turn, there is also some evidence that attempts to suppress an unwanted memory indeed momentarily prevent such memory-specific reactivation (*Detre et al., 2013*; *Gagnepain et al., 2014*; *Liu et al., 2016*; *Wimber et al., 2015*).

We have thus gained an evolved understanding of the mechanisms that are engaged transiently *during* the suppression of unwanted memories. By contrast, there is little evidence for the sustained neural *after-effect* of this process: Why do previously suppressed memories remain difficult to recall? Suppression has been argued to deteriorate the memory's neural representation (*Anderson et al., 2004*; *Depue, 2012*). Here, we test the hypothesis that it thus compromises later reactivation, even when one then tries to intentionally recall that memory (see also *Poppenk and Norman, 2014*). A deficient reactivation of cortical representations would hinder such recall attempts and diminish the vividness of the recollection.

To test this hypothesis, we conducted an fMRI study using an adapted *Think/No-Think* procedure (*Anderson and Green, 2001*; *Küpper et al., 2014*). First, participants learned associations between neutral objects (cues) and aversive scenes (target memories) (*Figure 1a*). During the suppression phase, they were then scanned by fMRI while they again encountered the cues. In this phase, participants were repeatedly prompted to recall the associated target for one-third of the cues (*recall condition*), whereas they were requested to prevent the retrieval of the targets for another third of the cues (*suppress condition*). In the suppress condition, we instructed participants to remain focused on the cue while trying to block out all thoughts of the accompanying target memory without engaging in any distracting activity (*Benoit and Anderson, 2012*; *Bergström et al., 2009*). Importantly, the remaining third of the cues were not presented during this phase (*baseline condition*). These cues and their associated targets thus serve as a baseline for the passive fading of memories that simply occurs due to passage of time (i.e. without any active suppression attempts).

To assess the degradation of neural memory representations over time, we also had participants recall each target in response to its cue both before and after the suppression phase. During these pre- and post-tests, they reported the vividness of the recalled memories. We thus assessed the phenomenological quality of the memories at the same time that we probed their neural reinstatement. Finally, participants engaged in a separate task that allowed us to train a pattern classifier to detect neural reactivation of complex and aversive scenes.

We tested our hypothesis by tracking the impact of suppression on neural reactivation, both distributed across the brain and more regionally specific in the PhC (see also *Detre et al., 2013*; *Gagnepain et al., 2014*; *Wimber et al., 2015*). Due to the strong association of the PhC with the processing of scenes, we considered it particularly sensitive to sustained disruptions of their neural representations. However, we do not suggest that suppression solely affects this region. In *Appendix 1—table 5*, we also report exploratory analyses of suggested candidate control regions, i.e. the amygdala, V1, the precuneus and the angular gyrus.

Specifically, we tested four key predictions. First, we expected that suppression would be associated with reduced scene reactivation. Second, we predicted that this effect would not be confined to the transient moment of active suppression but also linger on - as indexed by lower post-test reactivation

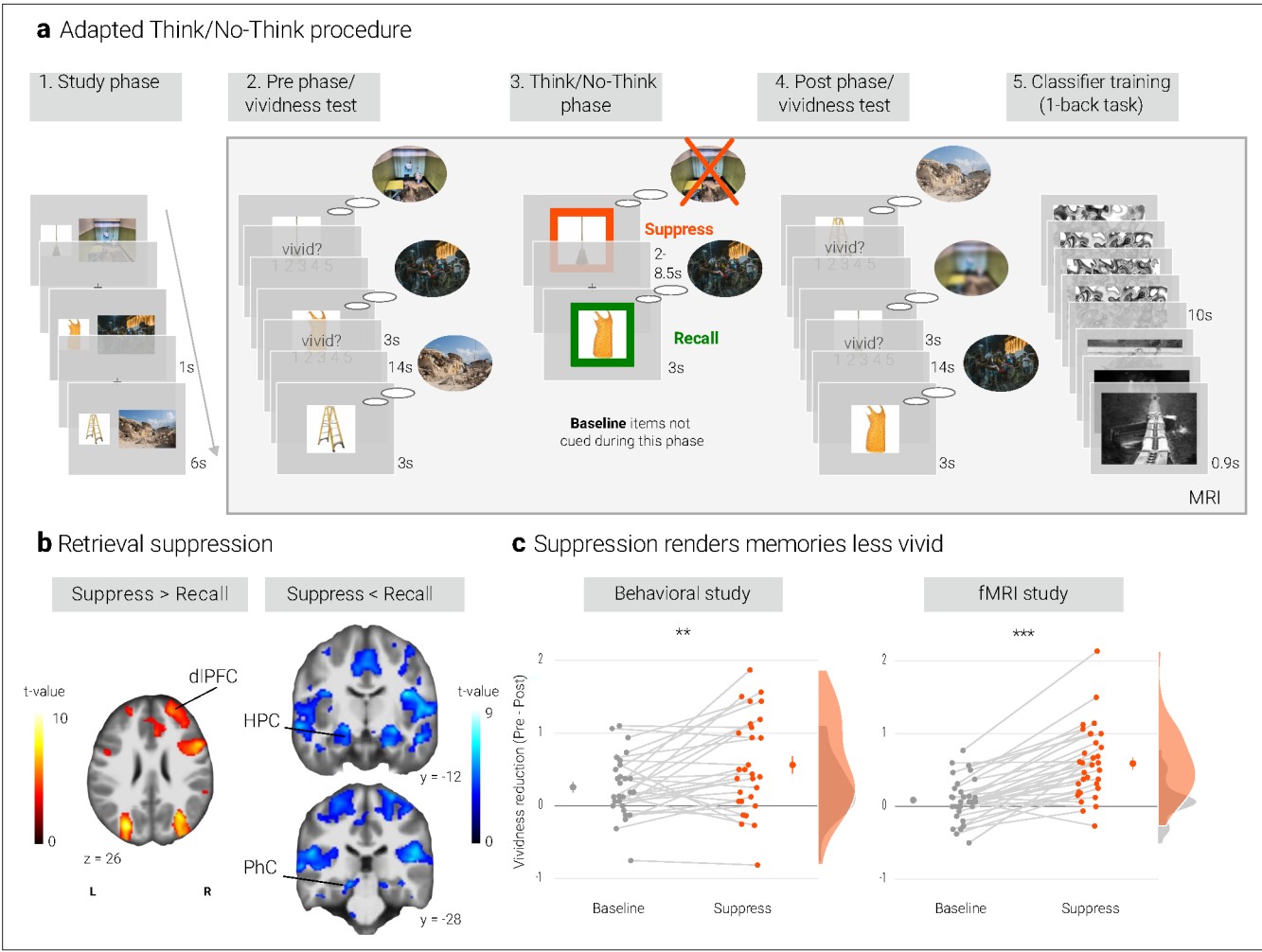

**Figure 1.** Experimental procedure, univariate MRI, and behavioral results. (**a**) Illustration of the adapted Think/No-Think procedure. Participants studied associations between unique objects and aversive scenes. Both during a pre- and a post-test, they covertly recalled all the scenes in response to the objects and rated the vividness of their recollection. In between these two tests, they performed the Think/No-Think phase. Specifically, for objects presented in a green frame, participants repeatedly recalled the associated scene (*recall condition*). By contrast, for objects presented in a red frame, they suppressed the retrieval of the associated scene (*suppress condition*). Note that we did not present a third of the objects during this phase (*baseline condition*). Afterwards, participants performed a one-back task that served to train a pattern classifier in detecting evidence for scene reactivation. Finally, participants once more recalled the memories outside the MRI scanner (now shown in this figure). The complete experimental procedure is comprehensively described in the methods. (NB) Following the IAPS user agreement, we have replaced the original pictures with similar scenes for this figure. In the original stimulus set, each object cue also features in its paired scene. (**b**) The suppression phase yielded the typical activity pattern associated with retrieval suppression, including greater activation in the right dorsolateral prefrontal cortex (dlPFC) and reduced activation in the hippocampus (HPC) and parahippocampal cortex (PhC) during *suppress* versus *recall* trials. For display purposes, the images are thresholded at *p* < .001, uncorrected, with a minimum cluster size of 50 voxels. (**c**) Suppression caused a reduction in self-reported vividness from the pre- to post-test that exceeded any change due to the passage of time as indexed by the *baseline* condition. This effect replicated across the fMRI study (*n* = 33) and a behavioral study (*n* = 30) with an independent sample. Large dots indicate the mean, error bars the standard error of the mean. *** *p* < .001, ** *p* < .01, * *p* < .05.

of previously suppressed memories. Third, in addition to a reduced reactivation of general scene information, we also predicted a weaker PhC reinstatement of the neural representations that are unique to the individual memories. Finally, if weaker neural reactivation constitutes the basis for the sustained suppression-induced reductions in vividness, we expected a relationship between these effects.

## Results

### Preventing retrieval yields the typical pattern associated with memory suppression

We first sought to establish whether our procedure elicited the activation pattern that has consistently been associated with retrieval suppression (e.g. *Anderson et al., 2004*; *Benoit and Anderson, 2012*; *Depue et al., 2006*; *Mary et al., 2020*). Suppressing versus recalling an aversive scene indeed led to increased activation in a number of brain regions including the right dlPFC and reduced activation in, amongst others, the bilateral hippocampi and PhC (*Figure 1b*, *Appendix 1—table 3*). This pattern is consistent with the engagement of the mechanism thought to mediate retrieval suppression (*Benoit and Anderson, 2012*; *Gagnepain et al., 2014*).

In the following, we test the hypothesis that this mechanism impairs subsequent retrieval attempts by hindering reinstatement of the neural memory representation. We thus examine suppression - induced changes in the phenomenological quality of the memories and their neural basis. These analyses focus on the critical comparison of the *suppress* versus *baseline* conditions. In the appendix, we explore possible effects of retrieval practice (*Karpicke and Blunt, 2011*; *Karpicke and Roediger, 2008*; *Roediger and Butler, 2011*) that is, contrasts of the *baseline* versus *recall* conditions.

### Suppression renders memories less vivid

We assessed the impact of suppression on the phenomenological quality of the memories by examining their change in vividness from the pre-test to the post-test. Indeed, there was a greater reduction for *suppress* than *baseline* memories as indicated by a significant interaction between time of test (pre, post) and condition (*baseline, suppress*) ($F(1, 32) = 46.18$, $p < .001$, $\eta^2 = 0.034$). However, the main effects of time of test ($F(1,32) = 28.87$, $p < .001$, $\eta^2 = 0.063$) and condition ($F(1,32) = 4.22$, $p = .048$, $\eta^2 = 0.007$) were also significant. Follow-up tests showed that suppression reduced the vividness of the memories ($t(32) = 6.60$, $p < .0001$, $d = 1.17$), whereas there was only a trend for baseline memories to change over time ($t(32) = 1.79$, $p = .08$, $d = 0.32$) (*Figure 1c*).

We obtained a similar pattern in a behavioral study with an independent sample (*Figure 1c*): again, the critical interaction between time of test and condition was significant ($F(1, 28) = 8.85$, $p = .006$, $\eta^2 = 0.015$). Additionally, there were main effects of time of test ($F(1, 28) = 21.78$, $p < .001$, $\eta^2 = 0.101$) and of condition ($F(1, 28) = 8.02$, $p = .008$, $\eta^2 = 0.02$). Baseline and suppress memories did not differ on the pre-test ($t(28) = 0.48$, $p = .63$, $d = 0.09$) but on the post-test ($t(28) = 3.26$, $p = .003$, $d = 0.62$). The follow-up tests again showed a reduction in vividness for the suppressed memories ($t(28) = 4.63$, $p < .0001$, $d = 0.88$). However, this time, the smaller reduction for baseline memories was also significant ($t(28) = 3.41$, $p = .002$, $d = 0.64$).

Consistent with prior research (*Stramaccia et al., 2021*), suppression thus had a replicable, detrimental impact on people's ability to vividly recall the suppressed memories. Importantly, we assessed the phenomenological quality of the memories during exactly those retrieval attempts that also provide the basis for our critical fMRI analyses. That is, in the following, we examine not only whether there is less reactivation of a memory during suppression (*Detre et al., 2013*; *Gagnepain et al., 2014*), but also the hypothesis that this effect then lingers on during these subsequent recall attempts.

### Univariate suppression-induced changes in brain activity

Before turning to scene-specific neural representations, we first probe whether suppression also yielded any generic after-effects in univariate brain activation. Specifically, we tested whether any voxels showed a stronger pre-to-post decrease for the suppressed than the baseline memories. Only a few regions showed such an effect, including parts of bilateral hippocampus (see *Appendix 1—table 4*).

However, to address our hypothesis, we move beyond univariate activation levels. By using multivoxel pattern analyses (*Norman et al., 2006*), we track changes in fine-grained activity patterns that are more specifically associated with the processing of scenes and with the reinstatement of individual memories.

## Establishing a linear classifier to detect scene reactivation

Memory retrieval reactivates the perceptual and conceptual representations elicited during encoding (*Dijkstra et al., 2020*; *Linde-Domingo et al., 2019*). To quantify the degree of such reactivation on a given trial, we trained a linear support vector machine (*Hebart et al., 2014*) on data from an independent task that participants had performed at the end of the MRI session. Specifically, the classifier learned to distinguish brain states associated with the perception of intact aversive scenes (similar to the ones used in the main task) versus morphed versions of the scenes. The morphed scenes were created via a diffeomorphic transformation that renders them unrecognizable while preserving their basic perceptual properties. Compared to conventional methods, such as scrambling, morphing has been shown to elicit neural activation that is more similar to activation induced by intact images (*Stojanoski and Cusack, 2014*).

Given the widespread nature of memory representations (*King et al., 2015*; *Rissman and Wagner, 2012*; *Ritchey et al., 2013*), we sought to test for global reactivation by training a classifier on all voxels of the respective participant's grey matter mask. Using cross-validation, the classifier reached a mean accuracy of 80.3% (*SD* = 17.4) on the training data, corroborating that it was able to distinguish brain states associated with the presentation of intact versus morphed aversive scenes ($t(32) = 10$, $p <$ .001). We then used the trained classifier to analyze the activity patterns on each trial of our memory tasks. Specifically, for a given trial, we calculated the dot product of the trial's activation map and the classifier's weight pattern (*Chang et al., 2015*; *Woo et al., 2017*). We take the resulting values to index the degree of scene reactivation (*Figure 2a*). (To ensure that the effects obtained with this mask are not simply driven by the PhC), we also ran all analyses for an additional ROI that excluded this region from the whole-brain mask. The results were virtually identical to the ones reported throughout the manuscript as described in the R Markdown available at OSF (https://osf.io/swxtd/?view_only=27da0e7814d24c3fafecddc2ab0a1163).

We also sought to test for more localized reactivation of scene information in the PhC, given the preferential engagement of this region for scene memories (*Epstein et al., 2003*; *Staresina et al., 2011*). Toward this end, we manually traced the parahippocampal cortices on each individual anatomical scan (*Insausti et al., 1998*; *Pruessner et al., 2002*; *Staresina et al., 2011*; *Figure 2*) and trained classifiers separately for the masks from the left and right hemisphere. These classifiers reached average cross-validation accuracies of 77.3% (*SD* = 16.6; $t(32) = 9.5$, $p <$ .001) and 82.6% (*SD* = 16.6; $t(32) = 11.3$, $p <$ .001), respectively.

We further validated our approach by examining the correspondence between the reactivation scores in the PhC and the vividness with which the memories could be recalled. The analysis was conducted on data from the pre-test. Because memories at that stage are still unconfounded by possible effects of the subsequent experimental manipulation, this allowed us to compute correlations based on all trials across the three conditions. Specifically, we correlated the scene reactivation and vividness scores for each participant and then performed one-sample t-tests on the individual Fisher-transformed correlation coefficients. These analyses showed that greater scene reactivation was indeed associated with more vivid recollections in left (*M* = 0.09, *95%* CI = [0.03 0.16], $t(32) =$ 2.82; $p =$ .01) and right PhC (*M* = 0.06, *95%* CI = [0.01 0.12], $t(32) = 2.27$; $p =$ .03).

## Reduced scene reactivation *during* suppression

The previous section established that the classifier provides a measure for the reactivation of scene information. We first examined whether such reactivation is reduced while participants intentionally try to suppress rather than to recall a memory. This was the case globally across the brain as indicated by the analysis based on the grey matter mask ($t(32) = 7.04$, $p <$ .001, $d = 1.22$).

For the PhC, a rANOVA with the factors hemisphere (left, right) and condition (recall, suppress) revealed an interaction of hemisphere and condition ($F(1,32) = 30.04$, $p <$ .001, $\eta^2 = 0.003$). Follow-up tests showed reduced scene evidence locally in the left ($t(32) = 2.84$, $p =$ .01, $d = 0.50$), though not right PhC ($t(32) = -0.60$, $p =$ .56, $d = -0.11$) (*Figure 2b*). These data suggest that participants were successful at controlling the retrieval of unwanted memories. At the same time, they further validate the use of the classifier as a measure of memory reactivation.

## Reduced global scene reactivation *following* suppression

Suppressed scenes were recalled less vividly than baseline scenes. We had hypothesized that this suppression-induced decline of the memories reflects a sustained reduction in the potential to

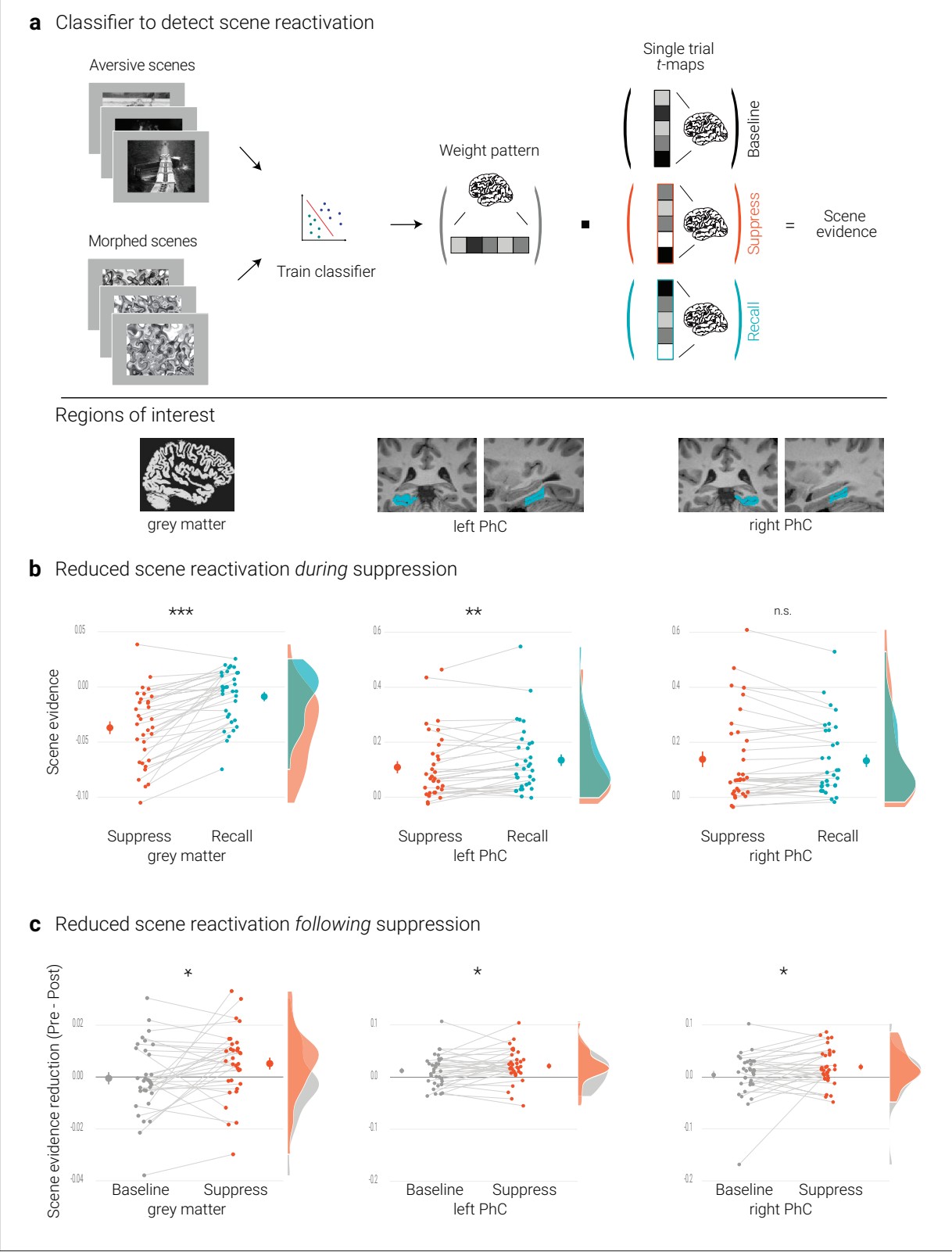

**Figure 2.** Effects of suppression on scene reactivation. (**a**) A linear support vector machine was trained on data of an independent task to discriminate neural activity patterns associated with the perception of intact versus morphed aversive scenes. (NB) Following the IAPS user agreement, we have replaced the original pictures with similar scenes for this figure. The dot product of the resulting weight pattern and single-trial t-maps was used as a proxy for reactivated scene information. We compared such scene evidence between conditions globally across the grey matter and more locally in

*Figure 2 continued on next page*

*Figure 2 continued*

the left and right parahippocampal cortices (PhC, manually segmented on the individual structural images). (**b**) Across the grey matter and locally in left PhC, there is less scene evidence while participants suppress than recall scene memories. (**c**) The suppression-induced reduction in scene evidence lingers on after suppression: scene evidence decreases from the pre- to the post- test for suppressed memories but not for baseline memories. This was the case across the grey matter and in the PhC. Larger dots indicate the mean, error bars the standard error of the mean. *** $p < .001$, ** $p < .01$, * $p < .05$, n = 33.

reactivate their neural representations. We thus expected reactivation scores for *suppress* memories to decline from the pre-test to the post-test to a larger degree than for *baseline* memories.

We tested for this effect by conducting a rANOVA on the global reactivation scores with the factors time of test (pre, post) and condition (suppress, baseline). This analysis yielded the expected significant interaction ($F(1,32) = 5.14$, $p = .03$, $\eta^2 = 0.006$), reflecting diminished scene reactivation for suppressed ($t(32) = 2.26$, $p = .03$, $d = 0.4$) but not for baseline memories ($t(32) = –0.2$, $p = .84$, $d = –0.03$) (*Figure 2c*).

## Reduced parahippocampal scene reactivation *following* suppression

As predicted, suppression also led to a sustained reduction of local scene reactivation in the PhC. This was corroborated by a rANOVA with the factors time of test (pre, post), condition (baseline, suppress), and hemisphere (left, right) that yielded the significant interaction between time and condition ($F(1,32) = 4.33$, $p = .046$, $\eta^2 = 0.003$) (in addition to a main effect of time, $F(1,32) = 8.83$, $p = .006$, $\eta^2 = 0.017$). This effect reflected the expected reduction in scene reactivation for suppressed ($t(32) = 3.77$, $p < .001$, $d = 0.67$) but not for baseline memories ($t(32) = 1.38$, $p = .18$, $d = 0.24$) (*Figure 2c*).

## A link between suppression-induced reductions in scene reactivation and vividness

Activity in the PhC has previously been associated with the number of details (*Qin et al., 2011*; *Tendolkar et al., 2008*) and the vividness (*Kensinger et al., 2011*; *Sheldon and Levine, 2013*; *Todd et al., 2013*) with which scenes can be recalled. We similarly observed that the recall of more vivid memories is accompanied by greater evidence for scene reactivation.

We accordingly hypothesized that a greater suppression-induced reduction in scene reactivation would lead to a greater reduction in vividness. We examined this hypothesis by exploiting the natural variation in people's ability to control unwanted memories.

For each participant, we quantified the suppression-induced reductions in vividness as the change from the pre- to the post-test for suppressed memories, corrected for by the change in vividness for baseline memories:

[1] suppression-induced reduction = (pre$_{suppress}$-post$_{suppress}$) – (pre$_{baseline}$ - post$_{baseline}$)

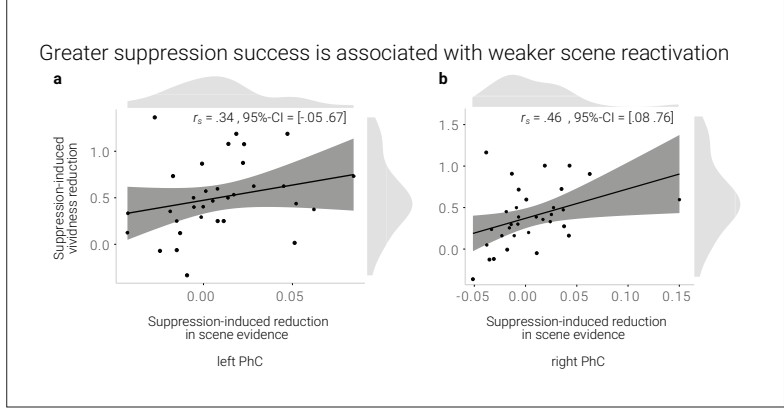

Greater suppression success is associated with weaker scene reactivation

**Figure 3.** A greater suppression-induced reduction in vividness is associated with a greater suppression-induced reduction in scene evidence in right PhC as indicated by a robust skipped Spearman's correlation. The left PhC showed a non-significant trend only for this effect. Black lines indicate linear regression lines, dark grey shades indicate 95%- confidence intervals, PhC: parahippocampal cortex.

We thus obtained an index of the deterioration in vividness that exceeds any effects that simply occur due to the passage of time (*Anderson et al., 2004*; *Benoit and Anderson, 2012*). Analogously, we calculated the degree of suppression-induced reductions in scene reactivation by subtracting the change score of the baseline memories from the score of the suppressed memories.

If the reduction in reactivation is linked to the reduction in vividness, we expected a positive correlation between the behavioral and neural suppression-induced reduction scores. Indeed, using robust skipped Spearman's correlations, we found a significant effect for the right ($r$ = 0.46, 95% CI = [0.08 0.76]) and a trend for the left PhC ($r$ = .34, 95% CI = [–0.05 0.67], *Figure 3*).

Taken together, suppression led to a sustained reduction of scene information on a global and local level. Moreover, the degree of reduced reactivation of scene information in the right PhC was linked to the decline of the memories' vividness.

To examine the origin of this lasting after-effect in the PhC, we further explored whether a stronger sustained suppression-induced reduction in PhC scene evidence is preceded by a stronger transient deactivation during suppression. However, there was no evidence for such a relationship across participants (left PhC: $r$ = 0.15, 95% CI = [–0.21 0.55]; right PhC: $r$ = 0.16, 95% CI = [–0.27 0.41]; skipped Spearman's correlations).

In contrast to the PhC, none of our candidate control regions showed sustained reductions in scene evidence following suppression (as detailed in *Appendix 1—table 5*) (all $F$ < 3.29; all $p$ > .08). Furthermore, only for the angular gyrus and the precuneus did we observe a trend for a correlation between the reductions in scene evidence and in vividness (angular gyrus: $r$ = 0.49, 95% CI = [–0.04 0.67], precuneus: $r$ = 0.32, 95% CI = [–0.05 0.61]; skipped Spearman's correlations). These results support the notion that the PhC may be particularly sensitive to suppression-induced changes in memory representations.

The data suggest that reduced PhC reactivation reflects the failure to retrieve scene features that would have made the recollections more vivid. In the following, we further examine this interpretation by assessing changes in the neural reinstatement of individual memory representations.

## Suppression success is associated with weaker memory-specific PhC pattern reinstatement

The classifier results indicate that suppression hinders the subsequent reactivation of scene information. However, they do not address the question whether this effect reflects reduced reinstatement of information that is specific to a particular memory. In a next step, we thus used Representational Similarity Analysis (RSA) (*Kriegeskorte et al., 2008*; *Nili et al., 2014*) to examine the reinstatement of activity patterns that are unique to the individual memories. We focus this analysis on the PhC, where the neural reinstatement of a particular memory should yield a unique and replicable activity pattern (*Martin et al., 2013*; *Staresina et al., 2012*; *Staresina et al., 2013*). Specifically, we expected a similar activity pattern to emerge whenever participants recall the same scene memory.

We quantified similarity by computing the Pearson correlation (*Kriegeskorte et al., 2008*; *Nili et al., 2014*) of the activity patterns across the pre- and post-test. As an index of memory-specific reinstatement, we then compared the similarity of a memory with itself (*same-item similarity*) and the similarity of a memory with all other memories of the same condition (e.g. baseline) (*different-item similarity*) (*Nili et al., 2020*; *Paulus et al., 2020*; *Staresina et al., 2012*; *Figure 4a*).

However, we note that the scene memories were probed with the same objects on both the pre- and post-test. Although the PhC is more sensitive to scene than object information (*Staresina et al., 2011*), any difference in same- versus different-item similarity may thus partly reflect the repetition of these retrieval cues. Critically, this caveat would not explain any differences in reinstatement for baseline versus suppressed memories.

A rANOVA with the factors scene identity (same, different), condition (baseline, suppress), and hemisphere (left, right) yielded the main effects of identity ($F(1,32)$ = 13.64, $p$ < .001, $\eta^2$ = 0.005) and hemisphere ($F(1,32)$ = 5.41, $p$ = .027, $\eta^2$ = 0.022) though not the critical interaction between scene identity and condition ($F(1,32)$ = 0.46, $p$ = .501, $\eta^2$ < 0.001). In follow-up analyses, we observed greater same- than different-item similarity for the baseline memories ($t(32)$ = 3.25, $p$ = .003, $d$ = 0.58), but only a non-significant trend for a numerically smaller effect for the suppressed memories ($t(32)$ = 1.84, $p$ = .075, $d$ = 0.32) (*Figure 4b*). The data thus provide some evidence for the replicable reinstatement of neural representations that are unique to the individual memories. Although there

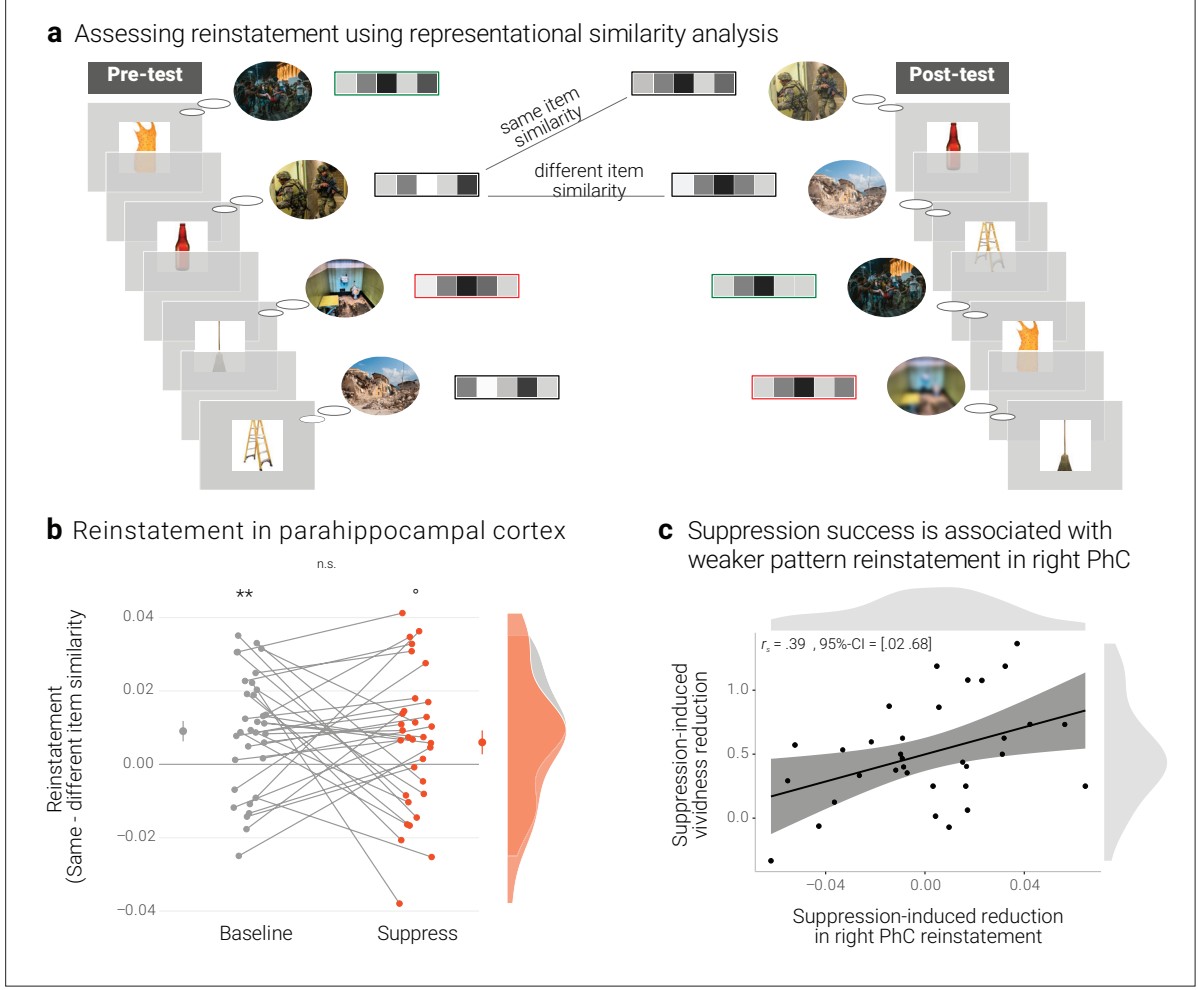

**Figure 4.** Effects of suppression on individual memory representations. (**a**) We estimated the reinstatement of the neural memory representations by assessing the similarity of the activity patterns across the pre- and the post-test. We take the difference between the same-item and different-item similarity as an index of neural reinstatement. (NB) Following the IAPS user agreement, we have replaced the original pictures with similar scenes for this figure. (**b**) In the PhC, the difference between same- and different-item similarity was significant for the baseline memories. These memories thus seem to have been consistently reinstated across the two tests. By contrast, the suppressed memories only showed a trend for this effect, though the critical interaction was not significant. Large dots indicate the mean, error bars the standard error of the mean. (**c**) A greater suppression-induced reduction in vividness was associated with a greater suppression-induced reduction in reinstatement in the right PhC (as indicated by a robust skipped Spearman's correlation). Black lines indicate linear regression lines, dark grey shades indicate 95% - confidence intervals, PhC: parahippocampal cortex. ** $p < .01$, * $p < .05$, ° $p < .1$, $n = 33$.

was only a trend for this effect in suppressed memories, it was - overall - not significantly smaller than for the baseline memories.

We had hypothesized that individuals who were more successful at suppression (as indicated by a greater reduction in vividness) should show evidence for a greater decline in neural reinstatement. As for the reactivation scores above, we thus examined the association between the suppression-induced reduction in vividness and the suppression-induced reduction in reinstatement. The latter was computed as:

[2] reinstatement = $r_{same-item}$ - $r_{different-item}$

[3] suppression-induced reduction = reinstatement$_{baseline}$ - reinstatement$_{suppress}$

Thus, a greater value indicates a greater reduction in memory-specific reinstatement. Mirroring the results of the pattern classifier, the skipped Spearman's correlation between the behavioral and neural effects was significant in the right ($r = 0.39$, 95% CI = [0.02 0.68]) though not left PhC ($r = 0.02$, 95% CI = [–0.37 0.41]) (**Figure 4c**).

As detailed in *Appendix 1—table 5*, the difference in same- versus different-item similarity was also not modulated by suppression in any of the candidate control regions. Unlike the PhC, neither of these regions showed a relationship between the degree of such modulation and suppression-induced reductions in vividness.

## Discussion

Research over the last two decades has demonstrated that we are able to control our unwanted memories by intentionally suppressing their retrieval (*Anderson and Green, 2001*; *Anderson and Hulbert, 2021*). This process weakens the avoided memory and can eventually lead to forgetting (*Küpper et al., 2014*; *Stramaccia et al., 2021*). Although this research has made strides in eluci-dating the transient mechanisms engaged *during* retrieval suppression (*Anderson and Hanslmayr, 2014*; *Benoit et al., 2015*; *Benoit et al., 2016*; *Benoit and Anderson, 2012*; *Depue et al., 2007*; *Gagnepain et al., 2014*; *Gagnepain et al., 2017*; *Mary et al., 2020*; *Paz-Alonso et al., 2013*), there is little evidence for the neural consequences that underlie the sustained subsequent changes in the retrievability of a suppressed memory.

In this study, we sought to tie the sustained phenomenological weakening of a suppressed memory to its neural basis. Successful episodic memory retrieval entails the reinstatement of a memory's repre-sentation (*Frankland et al., 2019*; *Linde-Domingo et al., 2019*; *Tonegawa et al., 2015*; *Wing et al., 2015*; *Xue et al., 2010*). It is fostered by hippocampal processes that complete the neural pattern of the original experience (e.g. of a particular scene) from a partial pattern provided by an adequate retrieval cue (e.g. of an object that was also part of the scene) (*Knierim and Neunuebel, 2016*; *Liu et al., 2012*; *Neunuebel and Knierim, 2014*). This process leads to the cortical reinstatement of a memory across the regions that had been involved in its original encoding (*Grande et al., 2019*; *Horner et al., 2015*; *Rissman and Wagner, 2012*).

Reinstatement has been examined in humans using fMRI by exploiting the distributed pattern of activity across voxels as a proxy of a memory's neural representation. Successful retrieval (e.g. of a particular scene) has thus been shown to be accompanied by a reactivation of categorical information (e.g. scene information) that is both widely distributed (*Cooper and Ritchey, 2019*; *King et al., 2015*; *Rissman and Wagner, 2012*; *Wing et al., 2015*) and localized to specific brain regions (e.g. the PhC; *Martin et al., 2013*; *Staresina et al., 2012*).

In the current study, the degree of scene evidence on a given trial scaled with the vividness with which a memory could be retrieved. This extends prior evidence showing that activation, particularly in the PhC, is stronger when scenes are recollected more vividly and in greater detail (*Kensinger et al., 2011*; *Qin et al., 2011*; *Sheldon and Levine, 2013*; *Tendolkar et al., 2008*; *Todd et al., 2013*). Our data thus further validate the use of classifier evidence as a marker of memory reactivation. We expected that intentional attempts to prevent retrieval should lead to reduced scene reactivation. This was the case in the current study across the grey matter as well as for the left PhC (see also *Detre et al., 2013*; *Gagnepain et al., 2017*; *Liu et al., 2016*). The data thus suggest that participants successfully suppressed unwanted retrieval.

Suppression also had a sustained impact on the avoided memories: It diminished their vividness on a subsequent test. This finding adds to the extant literature by showing that suppression does not only affect the objective availability of a memory in an all-or-none fashion (*Stramaccia et al., 2021*). By reducing the vividness of the memories, it also gradually diminishes the subjective quality of the infor-mation that can be retrieved. Such graded forgetting can be beneficial, for example when it allows for the continued conscious access to an aversive past event while dampening its affective impact (*Norby et al., 2010*; *Visser et al., 2018*, see also *Cooper and Ritchey, 2019*; *Parks and Yonelinas, 2007*; *Richter et al., 2016a*, for dissociations of retrieval success versus vividness).

Critically, the sustained fading of a suppressed memory has been argued to result from a deteriora-tion of its neural representation (*Anderson and Hanslmayr, 2014*; *Depue, 2012*). This deterioration, in turn, would lower the potential of these representations to be reinstated on later retrieval attempts. To test this hypothesis, we tracked changes in the reactivation of suppressed representations.

The classifier analysis yielded evidence for a greater reduction of scene reactivation for suppressed than for baseline memories. This was the case at a global level across the grey matter as well as locally in the PhC. Moreover, consistent with the contribution of the PhC to mnemonic vividness (*Todd et al., 2013*), participants who displayed a greater suppression-induced reduction in scene

reactivation in the right PhC also experienced a greater reduction in vividness. (Note that we do not suggest that it reflects a lateralized effect, given that the analogous analysis for the left PhC showed a similar, albeit non-significant, pattern.) Indeed, disruptions of cortical memory representations, such as in the PhC, may particularly lead to these graded effects of forgetting (*Gagnepain et al., 2014*). That is, as cortical representations get progressively weakened, they may become increasingly susceptible to interference from overlapping representations of similar memories (*Andermane et al., 2021*).

By comparison, hippocampal representations are encoded in a more orthogonal fashion (*Knierim and Neunuebel, 2016*; *Kumaran et al., 2016*) and may thus be largely protected from interference. The disruption of these representations would thus not manifest as graded forgetting but eventually lead to holistic forgetting in an all-or-none fashion (*Andermane et al., 2021*; *Horner et al., 2015*; *Richter et al., 2016b*). Indeed, a previous study did not obtain evidence for lingering effects of suppression on hippocampal reinstatement when the memories could still be recalled (*Liu et al., 2020*).

A pattern classifier is a powerful tool for inferring the reactivation of category-specific neural representations (*Haynes, 2015*; *Kuhl et al., 2012*; *Norman et al., 2006*; *Polyn et al., 2005*). As a downside, it does not provide evidence for the reinstatement of individual exemplars from within a category. Weaker scene evidence during the retrieval of suppressed memories may thus not only reflect reduced reactivation of the respective scene. Instead, it could conceivably also result from greater reactivation of additional, non-specific scene information during the retrieval of baseline memories. However, this interpretation is difficult to reconcile with the observation that, across participants, a greater suppression-induced reduction in scene reactivation was associated with a stronger decline in vividness.

In a complementary set of analyses, we used RSA to track the reinstatement of individual memory representations. Specifically, given that the retrieval of a memory should reinstate its representation, we expected similar activity patterns to emerge for a given memory on the pre- and on the post-test (*Danker et al., 2017*; *Staresina et al., 2012*; *Wing et al., 2015*; *Xue, 2018*). This was the case for the baseline condition, where the activity patterns were more similar for the comparison of a memory with itself than with the other memories. By contrast, the reinstatement was not significant for suppressed memories. However, we did not obtain the critical evidence that it was weaker than the baseline effect.

The absence of a significant difference in our study might simply reflect lower power for the more fine-grained, condition-rich analysis of individual activity patterns than for the more generic classifier (*Nili et al., 2020*). It may also result from the repetition of the identical objects as retrieval cues during the pre- and post-tests. Although the PhC is more sensitive to scenes than objects (*Staresina et al., 2011*), the repeated objects could have contributed to the similarity of the activity patterns. This would have been the case in both the baseline and suppress condition. Such a common effect driven by the object repetitions may have obscured any condition-specific effect driven by differences in scene reinstatement.

However, the absence of an overall effect might also reflect the varying degrees to which participants were successful at suppression (see also *Appendix 1—table 1*). Indeed, right parahippocampal reinstatement was particularly affected in those people who also experienced the strongest decline in vividness. This finding is particularly noteworthy as it corroborates the association that we had obtained with the more general index of scene reactivation. Together, the two sets of analyses thus converge in their support of our hypothesis that suppression leads to a sustained reduction in neural reinstatement.

Although we did not obtain a similar pattern in any of the additional candidate control regions, we do not suggest that suppression of scenes solely affects memory representations in the PhC. Indeed, the whole-brain classifier yielded evidence for sustained after-effects also when excluding the contribution of this region. These results indicate that further regions are affected by suppression. Nonetheless, the data are consistent with a preferential involvement of the PhC in the processing of scenes – a feature that may make this region particularly sensitive to suppression-induced changes in memory representations.

Computational modelling suggests that suppression deteriorates memory representations via a targeted inhibition of the respective representation's strongest, that is, most active, features (*Gagnepain et al., 2014*; *Poppenk and Norman, 2014*). These simulations imply that neural representations

need to be at least partially reactivated to become liable to disruption (*Elsey et al., 2018*; *Lee et al., 2017*; *Sinclair and Barense, 2019*).

Indeed, during initial suppression attempts, unwanted memories often involuntarily start intruding into awareness (*Benoit et al., 2015*; *Hellerstedt et al., 2016*; *Levy and Anderson, 2012*; *Mary et al., 2020*), indicating that they were partly reactivated. Such intrusions then become less frequent over time with repeated suppression attempts. The decrease in intrusions has been associated with a mechanism of reactive inhibitory control that is mediated by an upregulation of the dlPFC and a negative top-down modulation of hippocampal activation (*Benoit et al., 2015*; *Levy and Anderson, 2012*). This inhibitory signal may be relayed via the thalamic reuniens nucleus (*Anderson et al., 2016*) and there is some evidence that it relies on GABAergic activity in the hippocampus (*Schmitz et al., 2017*).

This account is reminiscent of the non-monotonic plasticity hypothesis which proposes that memories get weakened if they are moderately activated – irrespective of any intention to forget (*Kim et al., 2014*; *Norman et al., 2007*; *Poppenk and Norman, 2014*; *Ritvo et al., 2019*; *Sinclair and Barense, 2019*; *Wang et al., 2019*). On a neurophysiological level, this effect is reflected in long-term depression (i.e. synaptic weakening) following moderate postsynaptic depolarization (*Bear, 2003*; *Ritvo et al., 2019*). Suppression may complement such a passive learning process with a top-down process mediated by the dlPFC. By disrupting hippocampal retrieval, the top-down process may keep the reactivation of a memory to a moderate activity level and thus render its representation amenable to synaptic weakening.

There are a number of limitations to this work. First, as predicted, we do provide evidence for sustained after-effects of suppression on neural reactivation. We suggest that these are caused by a progressive weakening of the memory representations over repeated suppression attempts. This is consistent with prior evidence that unwanted memory intrusions become less frequent over time (*Benoit et al., 2015*; *Gagnepain et al., 2014*; *Levy and Anderson, 2012*; *Mary et al., 2020*). However, our study was not designed to assess the occurrence of memory intrusions. It therefore does not provide corroborating evidence for a gradual decline.

Second, we focused on the impact of suppression on the declarative components of aversive memories. Future studies could further determine links between possible reductions in negative affect and in neural reinstatement by assessing subjective emotional experiences. Such a relationship may be particularly pronounced in other brain regions that code for salience or negative affect, such as the amygdala (*Gagnepain et al., 2017*).

Finally, we obtained evidence for both, the transient engagement of retrieval suppression and for the subsequent sustained reductions in memory reactivation. However, it remains a future challenge to causally tie the effectiveness of the former neural inhibitory process to the latter after-effect.

To conclude, the current study set out to examine the neural consequences of suppression that underlie the phenomenon of suppression-induced forgetting (*Anderson and Green, 2001*; *Depue, 2012*; *Stramaccia et al., 2021*). We demonstrated that suppression rendered memories less vivid and, at the same time, provided evidence that it hindered the reactivation of their neural representations. Notably, a weaker reinstatement of the memories was also associated with a greater reduction in vividness. We thus tie the sustained phenomenological changes induced by suppression to their neural basis.

## Materials and methods
### Participants
Thirty-seven right-handed volunteers participated in the MRI study. They were all drawn from the participant database of the Max Planck Institute for Human Cognitive and Brain Sciences, reported no history of psychiatric or neurological disorder, gave written informed consent as approved by the local research ethics committee, and were reimbursed for their time. Four participants were excluded either due to technical problems (2), non-compliance with the instructions as assessed by a post-experimental questionnaire derived from *Hertel and Calcaterra, 2005* (please see below) (1), or drop out (1). We thus included 33 participants in the analysis (age: $M$ = 24.85 y, $SD$ = 2.14 y; 17 female, 16 male). We had aimed for a final sample of 30 participants and thus recruited 37 participants in

anticipation of possible exclusions due to non-compliance or excessive movement. This target sample size was chosen to exceed previous studies on suppression and based on our behavioral study.

The behavioral study included an independent sample of thirty-two volunteers. Two of these were excluded due to non-compliance with the instructions. For another participant, vividness ratings from the pre-test are missing due to technical issues, so that their data also had to be excluded from the analysis. These exclusions resulted in a final sample of 29 participants (*M* = 23.83, *SD* = 1.79, 14 female, 15 male) for the analysis of the vividness data. Demographic information for both the MRI and behavioral samples is summarized in *Appendix 1—table 1*.

## Materials

The stimuli for the experimental procedure were taken from *Küpper et al., 2014*. They comprised 60 object-scene pairs: 48 critical pairs and 12 filler pairs. The scenes were negative images depicting aversive scenes (e.g. accidents, injury, disaster) and were originally selected from the International Affective Picture System (IAPS; *Lang et al., 2008*) and online sources. The objects were photographs of familiar, neutral objects taken from *Brady et al., 2008*. Specifically, each object was chosen to resemble an object that was also part of its paired scene. By this, *Küpper et al., 2014* intended to mimic the situation in which an everyday object can trigger the involuntary retrieval of an aversive experience that had featured a similar object (*Ehlers, 2010*; *Ehlers and Clark, 2000*). Moreover, the objects were chosen to be peripheral to the respective scene (e.g. a doll in the corner of a room) and to be not essential to its gist. Per se, they were thus not space-defining, a feature that has previously been associated with PhC activity (*Auger et al., 2012*).

Throughout the experiment, all images were presented on a grey background. The 48 critical pairs were divided into three sets that were matched on salience of the objects, as well as the emotional valence and arousing nature of the scenes (*Küpper et al., 2014*). Assignment of the sets to the three conditions was counterbalanced across participants.

The task design for training the pattern classifier was based on *Poppenk and Norman, 2014*. It included black and white photographs of five different categories: aversive scenes, neutral scenes, morphed scenes, objects, and fruits. The aversive scenes were different items taken from similar data-bases as the critical items (IAPS; EmoPicS [*Wessa et al., 2010*], GAPED [*Dan-Glauser and Scherer, 2011*], NAPS [*Marchewka et al., 2014*]). The pictures were chosen to resemble the ones used for the original task in terms of complexity, closeness, and the degree to which they include humans. We moreover ensured that they depict similar aversive themes (e.g. fight, accident, disaster, war, hospital, injury).

We created morphed pictures of the aversive scenes with the procedure described by *Stojanoski and Cusack, 2014*. The morphed pictures retain the low-level visual features of the original pictures while ensuring that their content can no longer be recognized. We placed the pictures of the objects and fruits on top of phase-scrambled versions of the scenes and thus ensured that the images had the same size and rectangular shape. All images for the classifier training were normalized with respect to their luminance using the procedure described by *Detre et al., 2013*. The experiment was presented using Psychtoolbox (*Brainard, 1997*; *Pelli, 1997*).

## Procedures

### Experimental design

We tested the impact of suppression on memory reinstatement using an adapted version of the Think/No-Think procedure developed by *Küpper et al., 2014*. This procedure entailed four phases: an initial study phase, a pre-test, the suppression phase, and a post-test. These were followed by a classifier training task in the scanner (MRI study only) and an additional memory task (see appendix). The entire session took around four hours. By repeating the same procedure in two independent studies, we could corroborate the replicability of the suppression-induced reduction in vividness.

During the initial study phase, participants encoded all object-scene associations. First, they saw pairs of objects and scenes that were positioned next to each other. Participants tried to intentionally encode the associations and, in particular, the scenes in as much detail as possible. Each pair was presented for 6 s followed by a 1 s inter-trial-interval (ITI). Following initial encoding, we presented each object as a cue and asked participants to indicate within 5 s, via button press, whether they could fully recall the associated scene. Once they had pressed the button, they had again 5 s to choose the

correct scene out of an array of three different scenes (all of which were drawn from the actual stimulus set). The correct object-scene pair was then presented as feedback. This procedure was repeated up to three times until participants had correctly identified at least 60% of the scenes. To facilitate learning, this phase was split into two parts, each with half of the object-scene associations. Finally, participants were again shown all objects and asked once more to indicate whether they could recall the complete scene without feedback.

Participants then moved to the MRI scanner. Here, they saw all pairs a last time for 1.5 s each with an 800ms ITI. The extensive learning regimen and this refresher immediately prior to the critical parts of the experiment ensured that participants had encoded strong associations and were able to vividly recall the scenes. However, it made it less likely that suppression would induce absolute forgetting rather than gradual fading of the memories (*Benoit et al., 2015*).

During the pre-test, we presented all 48 reminders on the screen for 3 s each. Participants were asked to covertly recall the associated scene in as much detail as possible for the duration of the whole trial. They then had 3 s to rate the vividness of their recollection on a scale from 1 (not vivid at all) to 5 (very vivid). We presented no feedback at this stage. The rating was followed by a long ITI of 14 s. With this long ITI, we optimized our ability to detect the activity pattern associated with the recollection of a given scene with little contamination of the subsequent trial (*Poppenk and Norman, 2014*). The order of trials was pseudorandomized with at most three objects from the same condition presented in a row.

The Think/No-Think phase consisted of five blocks. During a block, each object was presented two times for 3 s. A green frame around an object indicated the *recall* task. That is, here participants were asked to recall the associated scene as vividly as possible. By contrast, a red frame around an object indicated the *suppress* task. Here, participants were asked to engage a mechanism that we have previously shown to disrupt hippocampal retrieval (*Benoit et al., 2016*; *Benoit and Anderson, 2012*; *Gagnepain et al., 2014*; *Mary et al., 2020*). That is, they tried to avoid the associated scene from coming to mind while focusing on the object on the screen. If the scene were to intrude into their awareness, they had to actively push it out of their mind. Importantly, a third of the objects were not shown during this phase. These items served as baseline memories to assess weakening due to the mere passage of time. The ITIs were optimized with optseq (https://surfer.nmr.mgh.harvard.edu/optseq/) and ranged from 2 s to 8.5 s with a mean of 3 s. Participants received extensive training and feedback on this procedure on the filler memories prior to entering the scanner. Immediately following the suppression phase, participants performed the post-test. This phase was identical to the pre-test but with a different pseudorandom presentation order.

Finally, participants engaged in a classifier training task (modelled on *Poppenk and Norman, 2014*) to obtain a neural pattern associated with the perception of aversive scenes. We presented pictures of the five categories in separate task blocks. During each block, they saw ten different pictures of the given category for 900ms with a 100ms ITI. Six of these pictures were randomly repeated within each block, thus resulting in 16 trials. Participants had to indicate the occurrence of these repetitions via a button press to ensure that they attended to the stimuli. Performance on the task was high, with participants responding correctly on 97.6% ($SD$ = 2.83) of the trials. Each category was presented in six blocks (for 30 blocks in total) in a pseudorandom presentation order with no more than two blocks of the same category in a row and with 10 s inter-block-intervals. After scanning, participants completed an additional memory task (see appendix).

Participants also completed a number of questionnaires. To assess their compliance with the instructions, they rated if they had *intentionally* recalled the suppress items, either (i) during the presentation of the reminders or (ii) following their offset or (iii) if they had generally made an effort to recall scenes of the suppress condition. For each of these questions, they used a scale from zero (never) to four (very often). The sum of the three scores serves as a measure of task compliance (*Hertel and Calcaterra, 2005*). The mean compliance score in the final MRI sample was 0.67 ($SD$ = 0.96, range: 0–3). Following standard procedure (*van Schie et al., 2013*; see also *Liu et al., 2021*), this sample excluded the one participant with a score of four. In the behavioral study, non-compliance led to the exclusion of two participants.

Participants further indicated that they had engaged more frequently in direct suppression than in the alternate strategy of thought substitution (scale from zero (never) to four (always), mean difference: 2.12, $t(32)$ = 9.8, $p$ < .001). They also indicated that they successfully suppressed the scenes on about

75% of the suppress trials (scale from zero (0%) to four (100%); $M = 2.99$, $SD = 0.92$) and that they remembered the respective scenes on about 86% of the recall trials ($M = 3.44$, $SD = 0.88$). Together, participants thus indicated that they generally complied with the task instructions and that they were successful at employing retrieval suppression as a means of avoiding the unwanted memories.

Moreover, they filled in Beck's Depression Inventory (*Beck et al., 1996*), the Thought Control Ability Questionnaire (*Luciano et al., 2005*) and the State-Trait Anxiety Inventory (*Spielberger et al., 1983*). These features of our sample are summarized in *Appendix 1—table 1*.

### fMRI data acquisition

We used a 3T Siemens Prisma MRI Scanner with a 32-channel head coil at the Max Planck Institute for Human Cognitive and Brain Sciences. Structural images were acquired with a T1-weighted MPRAGE protocol (256 sagittal slices with interleaved acquisition, field of view = 240 mm by 176 mm, 1 mm isotropic voxels, TR = 2300ms, TE = 2.98ms, flip angle = 9°, phase encoding: anterior-posterior, parallel imaging = GRAPPA, acceleration factor = 2). Functional images were acquired using a whole brain multiband echo-planar imaging (EPI) sequence (field of view = 192 mm by 192 mm, 2 mm isotropic voxels, 72 slices with interleaved acquisition (angled 15° towards coronal from AC-PC), TR = 2000ms, TE = 25ms, flip angle = 90°, phase encoding: anterior-posterior, MF = 3) (*Feinberg et al., 2010*; *Moeller et al., 2010*). 369 volumes were acquired in pre- and post-tests, 197 volumes in each suppression block and 395 volumes in the classifier training. The first five volumes of each run were discarded to allow for T1 equilibration effects. Pulse oxymeter data were collected on participants' left hand. Participants gave their responses via a 5-button box with their right hand.

## Analyses

### fMRI data preprocessing

The MRI data were first converted into the Brain Imaging Data Structure (BIDS) format (*Gorgolewski et al., 2016*). All data preprocessing was performed using the default preprocessing steps of fMRIPrep 1.5.0rc2, based on Nipype 1.2.1. (*Esteban et al., 2018*): The respective T1 volume was corrected for intensity non-uniformity and skull-stripped, before it was segmented into cerebrospinal fluid (CSF), white matter (WM), and grey matter (GM). It was then spatially normalized to the ICBM 152 Nonlinear Asymmetrical template version 2009c using nonlinear registration.

The functional data were slice-time corrected, motion corrected, and corrected for susceptibility distortions using fMRIPrep's fieldmap-less approach. They were then coregistered to the corresponding T1 image using boundary-based registration with six degrees of freedom. Physiological noise regressors were extracted to allow for component-based noise correction. Anatomical CompCor components were calculated within the intersection of the subcortical mask and the union of CSF and WM masks, after their projection to the native space of each functional run. Framewise displacement was also calculated for each functional run. For further details of the pipeline, including the software packages used by fMRIPrep, please refer to the online documentation (https://fmriprep.org/en/20.2.0/). Our univariate analyses were performed in MNI space (following smoothing with a Gaussian kernel of 6 mm FWHM), whereas the multivariate pattern analyses (MVPA) were done on unsmoothed data in native space.

### Regions of interest

We manually segmented the PhC on the individual T1-weighted structural images, following the anatomical demarcation protocol by *Insausti et al., 1998* and *Pruessner et al., 2002*. Specifically, we defined the PhC as the posterior third of the parahippocampal gyrus (*Staresina et al., 2012*). We further used the individual grey matter masks, segmented using FSLfast (in the fMRIPrep pipeline), as an ROI.

We focused on the PhC given its prominent contribution to memories for scenes (*Bohbot et al., 2006*; *Horner et al., 2015*; *Staresina et al., 2013*; *Staresina et al., 2012*). However, we do not suggest that effects of suppression are solely attributable to changes to representations in this region. We also explored possible sustained effects of suppression in a number of suggested bilateral candidate control regions. These are the angular gyrus and precuneus, given their association with the phenomenological quality of memories (*Richter et al., 2016a*; see also *Brodt et al., 2018*), the amygdala, given its association with the processing of aversive stimuli (*Depue et al., 2007*; *Gagnepain*

*et al., 2017*), and V1, given the finding of transient suppression-reduced reductions in early visual cortex (*Depue et al., 2007*; *Gagnepain et al., 2014*). Masks of the amygdala, V1, and precuneus were taken from the brainnetome atlas (*Fan et al., 2016*), and of the angular gyrus from the AAL2 atlas (*Tzourio-Mazoyer et al., 2002*) given the lack of an adequate brainnetome region.

### First-level fMRI analysis
Data were analyzed using SPM12 (https://fil.ion.ucl.ac.uk/spm). We decomposed the variance in the BOLD time series using general linear models (GLM) (*Penny et al., 2011*). For the univariate analysis of the suppression phase, we analyzed the data with a GLM including a regressor for the trials of the recall condition and a regressor for the trials of the suppress condition.

For our multivariate pattern analyses (MVPA), we assessed the individual activity patterns adopting a least-squares-single approach (*Mumford et al., 2012*). That is, for the pre- and post-test, we estimated separate GLMs for each trial with a regressor for that specific trial and a second regressor for all other trials. For the suppression phase, a given GLM included a regressor coding for all repetitions of the same object and a second regressor for all other trials. For the classifier training task, we estimated separate GLMs for each block with a regressor for that specific block and a second regressor for all other blocks.

All these regressors coded for the respective 3 s of each trial (or 16 s of each block for classifier training) and were convolved with the canonical hemodynamic response function. In addition, each GLM included six head motion parameters, framewise displacement, the first six aCompCor components and a block regressor as nuisance regressors. We then applied a 128 Hz high-pass filter to the data and the model. For the MVPA analyses, the resulting parameter estimates were transformed into t-values via a contrast of the respective individual trial versus all other trials.

### Classification analysis
We performed the classifier analysis using the decoding toolbox (*Hebart et al., 2014*). Specifically, we trained a linear support vector machine for each participant to distinguish activity patterns associated with intact aversive scenes versus their morphed versions. We employed a leave-one-out cross-validation approach that used, on each iteration, eleven of the twelve blocks as training data. This procedure assigns a linear weight to each voxel that reflects its importance in discriminating the two classes, thus creating a weight map. We then used the transformed weight pattern (*Haufe et al., 2014*) to estimate reactivation as the degree of scene evidence during each trial of the pre-test, post-test, and suppression phase. This was done by calculating the dot product of the weight pattern and the respective individual t-map.

### Representational similarity analysis
We examined the reinstatement of unique memory representations using representational similarity analysis (RSA). Specifically, we assessed whether the retrieval of a given scene was associated with a similar neural activity pattern before and after the suppression phase. This analysis used the RSA toolbox (*Nili et al., 2014*). It was based on the 48 trials from the pre-test and the post-test. We computed the similarity values using Pearson correlation across all voxels of the respective ROI (*Kriegeskorte et al., 2008*). Specifically, we assessed the similarity of each item with itself (same-item similarity) and the average similarity of the item with all 15 other items from the same condition (different-item similarity) (*Nili et al., 2020*). By constraining the different-item similarity to items of the same category, we ensure that any differences with the same-item similarity do not simply reflect general condition differences (i.e. systematic pattern differences for baseline versus suppress items). The similarity estimates were then Fisher-transformed and averaged for each condition within subjects. We determined the magnitude of pattern reinstatement as the difference score between same-item and different-item similarity.

### Statistical analyses
Statistical tests were done with R version 4.0.3 (R Core Team, 2019). Repeated measures ANOVAs were conducted with the afex package (Type 3 sums of squares; *Singmann et al., 2020*) and effect sizes are reported as generalized eta squared. Follow-up tests were based on estimated marginal means (emmeans package, *Lenth, 2020*) using pooled variances and degrees of freedom (based on

**Table 1.** Software resources used.

| Designation | Source/ Reference | Identifier |
|---|---|---|
| R version 4.0.3, RStudio | R Project for Statistical Computing | https://www.r-project.org/ |
| *MATLAB, 2017* | Mathworks | https://mathworks.com/ |
| Psychtoolbox 3 | *Pelli, 1997* | https://psychtoolbox.org/ |
| FMRIPrep | Poldrack Lab, Standford University | https://fmriprep.org/en/1.5.0/ |
| SPM12 | Wellcome Centre for Neuroimaging, UCL | https://fil.ion.ucl.ac.uk/spm/ |
| The decoding toolbox | *Hebart et al., 2014* | https://sites.google.com/site/tdtdecodingtoolbox/ |
| RSA toolbox | *Nili et al., 2014* | https://github.com/rsagroup/rsatoolbox |
| ITK-SNAP | *Yushkevich et al., 2006* | http://itksnap.org/ |

the Welch–Satterthwaite equation). The significance level was set to 5%. The robust skipped Spearman's correlations were estimated in Matlab (*MATLAB Version 9.3.0.713579 (R2017b)*, 2017) using the robust correlation toolbox (*Pernet et al., 2012*). All software resources used are listed in *Table 1*.

## Acknowledgements

We thank Philipp Paulus for help in implementing the classifier training task, Roxanne Eisenbeis, Johanna Fiebig, Leonie Kanne, and Sarah-Lena Schaefer for assistance in data acquisition, data transcription and scoring, Philipp Paulus, Heidrun Schultz, Hanna Stoffregen, and Angharad Williams for comments on a draft of the manuscript, as well as Bernhard Staresina for advice on the manual PhC segmentations. A-KM and RGB were funded by a Max Planck Research Group awarded to RGB. The funder had no role in study design, data collection and interpretation, or the decision to submit the work for publication.

## Additional information

### Funding

| Funder | Grant reference number | Author |
|---|---|---|
| Max Planck Society | | Ann-Kristin Meyer, Roland G Benoit |

The funders had no role in study design, data collection and interpretation, or the decision to submit the work for publication.

### Author contributions

Ann-Kristin Meyer, Conceptualization, Data curation, Formal analysis, Investigation, Methodology, Project administration, Software, Validation, Visualization, Writing – original draft, Writing – review and editing; Roland G Benoit, Conceptualization, Formal analysis, Funding acquisition, Investigation, Methodology, Project administration, Resources, Supervision, Validation, Writing – original draft, Writing – review and editing

### Author ORCIDs

Ann-Kristin Meyer ⓘ http://orcid.org/0000-0001-7468-2911
Roland G Benoit ⓘ http://orcid.org/0000-0002-7127-4899

### Ethics

The Ethics committee of the Medical Faculty, University of Leipzig, Germany, approved the study (protocol number 333/16-ek). Participants provided written informed consent to participate in the study and for group results to be published in a scientific journal.

Decision letter and Author response
Decision letter https://doi.org/10.7554/eLife.71309.sa1
Author response https://doi.org/10.7554/eLife.71309.sa2

## Additional files

### Supplementary files
• Transparent reporting form

### Data availability
All data are publicly available that support the findings, i.e., that are depicted in the figures and that the inferential statistics are based on. These are provided as source data files (https://osf.io/swxtd/?view_only=27da0e7814d24c3fafecddc2ab0a1163). We additionally provide the statistical map of the univariate fMRI analysis (https://neurovault.org/collections/KAZGAACE/) as well as all custom code used to analyze the data (including a full R Markdown) (https://osf.io/swxtd/?view_only=27da0e7814d24c3fafecddc2ab0a1163). However, we can not openly share the raw, unprocessed MRI data, because the participants did not give consent for these data to be released publicly within the General Data Protection Regulation 2016/679 of the EU. The data will be stored on the servers of the Max Planck Institute of Cognitive and Brain Sciences and can be made available to individual researchers on informal request to the corresponding author.

The following dataset was generated:

| Author(s) | Year | Dataset title | Dataset URL | Database and Identifier |
|---|---|---|---|---|
| Meyer AK, Benoit RG | 2021 | Suppression weakens unwanted memories via a sustained reduction of neural reactivation | https://neurovault.org/collections/KAZGAACE/ | NeuroVault, 9388 |

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

# Appendix 1

**Appendix 1—table 1.** Demographic information.

| Study | Sample size | Age in years M(SD) | TCAQ | BDI | STAI-S | STAI-T |
|---|---|---|---|---|---|---|
| Behavioral Study | 30 (15 female) | 23.83 (1.76) | 81.61 (15.11) | | | |
| fMRI study | 33 (17 female) | 24.85 (2.14) | 87.94 (16.2) | 4.27 (5.06) | 34.78 (8.37) | 35.78 (8.87) |

Note: Of the questionnaire data, exploratory analyses only suggested a relationship between the TCAQ and suppression-induced reductions in vividness. There was a trend for a skipped Spearman correlation in the MRI study ($r = 0.32$, 95% CI = [–0.08 0.58]) and a significant correlation across the two studies ($r = 0.27$, 95% CI = [0.004 0.49]).

**Appendix 1—table 2.** Control regions: Suppress >Recall.

| Region | ~BA | Hemi-sphere | Voxels | x | MNI y | z | t-value |
|---|---|---|---|---|---|---|---|
| | 18 | L | | | -28 | -88 | 10 | 10.83 |
| | 19 | L | | | -46 | -80 | -8 | 10.26 |
| Occipital, inferior temporal | 19 | L | 10,034 | | -30 | -90 | 20 | 9.79 |
| | 18 | R | | | 40 | -90 | 4 | 9.03 |
| | 37 | R | | | 36 | -44 | -18 | 8.82 |
| Occipital, inferior temporal | 7 | R | 9,037 | | 30 | -68 | 30 | 8.68 |
| | 6 | L/R | | | 8 | 22 | 62 | 8.89 |
| | 13 | R | | | 28 | 20 | -12 | 8.59 |
| Superior & middle frontal * | 6 | R | 10,089 | | 48 | 6 | 28 | 7.84 |
| | | L/R | | | 2 | -6 | 18 | 6.54 |
| | | R | | | 4 | -22 | 18 | 5.48 |
| Thalamus | | R | 681 | | 12 | -14 | 18 | 5.07 |
| | 13 | L | | | -36 | 20 | -4 | 6.51 |
| | 13 | L | | | -30 | 20 | -12 | 6.51 |
| Inferior frontal | 45 | L | 723 | | -42 | 22 | 8 | 6.51 |
| | 6 | L | | | -48 | 4 | 30 | 6 |
| | 6 | L | | | -56 | 8 | 44 | 5.12 |
| Middle frontal | 6 | L | 738 | | -48 | 14 | 42 | 4.95 |
| | 47 | L | | | -34 | 46 | -14 | 4.88 |
| Orbitofrontal | 47 | L | 166 | | -46 | 44 | -14 | 4.5 |
| | 9 | L | | | -24 | 46 | 34 | 4.59 |
| | 10 | L | | | -24 | 50 | 26 | 4.59 |
| Superior & middle frontal | 10 | L | 219 | | -30 | 56 | 20 | 4.03 |

Note. Thresholded at $p < .05$, FWE cluster corrected, with a cluster forming threshold of $p < .001$; local maxima more than 8 mm apart, minimum of 10 voxels. BA = Brodmann area.
*includes peak for direct suppression by **Benoit and Anderson, 2012**

**Appendix 1—table 3.** Modulated regions: Suppress <Recall.

| Region | ~BA | Hemisphere | Voxels | x | y | z | t-value |
|---|---|---|---|---|---|---|---|
| Cuneus, precuneus, HPC, amygdala, PhC | 19 | L/R | | -10 | -82 | 34 | 9.88 |
| | 23 | L | | 12 | -56 | 24 | 9.85 |
| | 18 | L | 31,631 | -18 | -62 | 22 | 9.34 |
| | | L | | 12 | -52 | -62 | 5.59 |
| | | L | | 22 | -48 | -52 | 5.22 |
| Cerebellum | | L | 185 | 18 | -58 | -48 | 4.2 |

Note. Thresholded at $p < .05$, FWE cluster corrected, with a cluster forming threshold of $p < .001$; local maxima more than 8 mm apart, minimum of 10 voxels. BA = Brodmann area, HPC = Hippocampus, PhC = Parahippocampal cortex.

**Appendix 1—table 4.** Suppression-induced reductions in univariate brain activity: ($Baseline_{Post} > Suppress_{Post}$) > ($Baseline_{Pre} > Suppress_{Pre}$).

| Region | ~BA | Hemi-sphere | Voxels | x | y | z | t-value |
|---|---|---|---|---|---|---|---|
| Middle temporal, fusiform, inferior occipital | 37 | R | 290 | 54 | -70 | 2 | 6.17 |
| Precentral, Postcentral | 6 | L | 344 | -64 | -18 | 44 | 5.37 |
| | | | | -34 | -14 | 62 | 5.23 |
| | | | | -28 | -34 | 70 | 4.72 |
| Lingual | 18/19 | R | 132 | 20 | -76 | -4 | 5.22 |
| | | | | 26 | -70 | -4 | 4.18 |
| Supplementary motor area | 6 | L | 180 | -6 | -14 | 56 | 5.07 |
| | | | | -14 | -16 | 46 | 4.88 |
| | | | | -10 | -6 | 50 | 3.87 |
| Middle Temporal | 19 | L | 156 | -52 | -70 | 8 | 4.81 |
| | | | | -40 | -70 | 12 | 4.25 |
| Hippocampus | | R | 133 | 30 | -6 | -26 | 4.67 |
| | | | | 24 | -14 | 16 | 4.55 |
| | | | | 16 | -8 | -18 | 3.97 |
| Superior occipital | 19 | L | 126 | -18 | -84 | 34 | 4.59 |
| | | | | -12 | -84 | 34 | 4.59 |
| Lingual, cerebellum | 37/19 | R | 118 | 26 | -44 | -10 | 4.25 |
| | | | | 30 | -56 | -24 | 4.36 |
| | | | | 20 | -54 | -12 | 3.96 |

Note. Thresholded at $p < .05$, FWE cluster corrected, with a cluster forming threshold of $p < .001$; local maxima more than 8 mm apart, minimum of 10 voxels. BA = Brodmann area.

**Appendix 1—table 5.** Results of candidate control regions.

| ROI | Classifier accuracy M (SD) | Reduced scene reactivation *following* suppression | Correlation between suppression-induced reductions in scene reactivation and in vividness | Reinstatement: same-item similarity >different item similarity | Correlation between suppression-induced reductions in reinstatement and in vividness |
|---|---|---|---|---|---|
| Amygdala | 62.4% (21.2) $t_{(32)} = 3.6$, $p = .002$ | $M_B$ pre = –0.001 $M_B$ post = –0.003 $M_S$ pre = –0.003 $M_S$ post = –0.007 $F_{(1,32)} = 1.13$, $p = .29$ | $r = 0.06$, 95% CI = [–0.35 0.43] | $M_B$ same = 0.021 $M_B$ different = 0.018 $M_S$ same = 0.019 $M_S$ different = 0.022 $F_{(1,32)} = 0.34$, $p = .56$ | $r = –0.02$, 95% CI = [–0.38 0.35] |
| Angular gyrus | 76.8% (16) $t_{(32)} = 9.6$, $p < .001$ | $M_B$ pre = –0.030 $M_B$ post = –0.023 $M_S$ pre = –0–028 $M_S$ post = –0–021 $F_{(1,32)} = 0.0$, $p = .97$ | $r = 0.49$, 95% CI = [–0.04 0.67] | $M_B$ same = 0.146 $M_B$ different = 0.125 $M_S$ same = 0.132 $M_S$ different = 0.123 $F_{(1,32)} = 2.1$, $p = .16$ | $r = 0.1$, 95% CI = [–0.23 0.41] |

*Appendix 1—table 5 Continued on next page*

*Appendix 1—table 5 Continued*

| ROI | Classifier accuracy M (SD) | Reduced scene reactivation *following* suppression | Correlation between suppression-induced reductions in scene reactivation and in vividness | Reinstatement: same-item similarity >different item similarity | Correlation between suppression-induced reductions in reinstatement and in vividness |
|---|---|---|---|---|---|
| Precuneus | 80.6% (13.8) $t(32) = 12.8$, $p < .001$ | $M_B$ pre = 0.004 $M_B$ post = 0.008 $M_S$ pre = 0.007 $M_S$ post = 0.009 $F(1,32) = 0.15$, $p = .71$ | $r = 0.32$, 95% CI = [−0.05 0.61] | $M_B$ same = 0.103 $M_B$ different = 0.093 $M_S$ same = 0.108 $M_S$ different = 0.092 $F(1,32) = 2.1$, $p = .16$ | $r = −0.09$, 95% CI = [−0.42 0.23] |
| V1 | 70.7% (14.5) $t(32) = 8.2$, $p < .001$ | $M_B$ pre = −0.096 $M_B$ post = −0.089 $M_S$ pre = −0.088 $M_S$ post = −0.097 $F(1,32) = 3.29$, $p = .08$ | $r = 0.2$, 95% CI = [−0.21 0.58] | $M_B$ same = 0.524 $M_B$ different = 0.482 $M_S$ same = 0.542 $M_S$ different = 0.483 $F(1,32) = 3.94$, $p = .056$ | $r = 0.05$, 95% CI = [−0.21 0.58] |

Note. B = Baseline, S = Suppress. Robust skipped spearman correlations

## Retrieval practice

Repeatedly retrieving a memory facilitates subsequent recall attempts (*Karpicke and Blunt, 2011*; *Karpicke and Roediger, 2008*; *Roediger and Butler, 2011*). Here, we examine the effects of such retrieval practice by comparing the recall and baseline conditions.

### Retrieval practice preserves vividness

Memories of the recall condition seemed to remain more vivid than those of the baseline condition from the pre- to the post-test (*Appendix 1—figure 1a*). We tested this effect with a repeated measures ANOVA with the factors condition (recall, baseline) and time of test (pre, post). The effect of condition was significant ($F(1,32) = 5.38$, $p = .027$, $\eta^2 = 0.010$) and, critically, so was the interaction between condition and time of test ($F(1,32) = 5.53$, $p = .025$, $\eta^2 = 0.007$). Indeed, the conditions only exhibited a significant difference after ($t(32) = -2.90$, $p = .01$ $d = -0.51$) but not before ($t(32) = 0.32$, $p = .75$, $d = -0.06$) retrieval practice. Further follow-up tests revealed a non-significant trend for a decrease in vividness for baseline memories ($t(32) = 1.79$, $p = .08$, $d = 0.32$) with no evidence for a change for recall memories ($t(32) = -1.60$, $p = .12$, $d = -0.28$).

We obtained the same pattern in our behavioral study, with a significant effect of condition ($F(1,28) = 9.60$, $p = .004$, $\eta^2 = 0.028$) and a significant interaction between condition and time of test ($F(1,28) = 15.83$, $p < .001$, $\eta^2 = 0.014$). Again, conditions did not significantly differ on the pre ($t(28) = -0.9$, $p = .37$, $d = -0.17$) but on the post-test ($t(28) = -4.19$, $p < .001$, $d = -0.79$). Here, further follow-up tests indicated a decrease in vividness for baseline ($t(28) = 3.41$, $p = .002$, $d = 0.64$), but not for recall memories ($t(28) = 0.29$, $p = .77$, $d = -0.06$).

### Effects on scene reactivation

Successful retrieval is accompanied by reinstatement of the brain activity pattern that was present during encoding (*Ritchey et al., 2013*; *Wing et al., 2015*; *Xue et al., 2010*). However, there is mixed evidence how repeatedly retrieving a memory alters this neural representation. Some evidence indicates that it remains stable over time (*Ferreira et al., 2019*), whereas other studies note that it becomes more differentiated (Hulbert & Norman, 2015; *Karlsson Wirebring et al., 2015*; *Liu et al., 2020*; *Ye et al., 2020*). Globally across the whole brain, the change in scene evidence from the pre- to the post-test did not differ for recall versus baseline memories ($F(1,32) = 2.00$, $p = .167$, $\eta^2 = 0.001$).

By contrast, the analysis of the PhC data, with the additional factor hemisphere (left, right), yielded a significant interaction between time of test and condition ($F(1,32) = 9.25$, $p = .005$, $\eta^2 = 0.003$) (and also a significant main effect of time of test ($F(1,32) = 4.83$, $p = .035$, $\eta^2 = 0.016$)), a main effect of condition ($F(1,32) = 9.97$, $p = .003$, $\eta^2 = 0.006$) and an interaction between time of test and hemisphere ($F(1,32) = 4.9$, $p = .034$, $\eta^2 = 0.003$) (*Appendix 1—figure 1b*). The interaction of time of test and condition reflected a relative decrease in scene evidence for recall memories ($t(32) = 2.72$, $p = .01$, $d = 0.48$), with no evidence for a difference for baseline memories ($t(32) = 1.38$, $p = .18$, $d = 0.24$) (*Appendix 1—figure 1b*).

## Relationship between retrieval-practice effects on vividness and PhC scene reactivation

We further examined the relationship between the effects of retrieval practice on vividness and on scene reactivation in the PhC. We therefore derived, for each measure, an index of retrieval practice by subtracting the temporal difference score (pre - post) of the baseline condition from the difference score of the recall condition. On both of these indices, a more negative value thus indicates a greater practice effect. The two indices were indeed positively correlated for the left PhC as indicated by a skipped Spearman's correlation of $r = .34$, 95% CI = [0.01 0.59] (*Appendix 1—figure 1c*). People who showed a greater retrieval-induced increase in vividness yielded less of a reduction – or even an increase – in scene evidence (*Appendix 1—figure 1*). However, this effect was not present in the right PhC (skipped Spearman's correlation: $r = -0.07$, 95% CI = [-0.46 0.35]).

## Effects on reinstatement of individual representations

A rANOVA with the factors identity (same, different), condition (recall, baseline) and hemisphere (left, right), yielded evidence for overall significant pattern reinstatement ($F(1,32) = 16.49$, $p < .001$, $\eta^2 = 0.006$). This effect did not interact with condition ($F(1,32) = 0.03$, $p = .862$, $\eta^2 < 0.001$). Notably, similar to the classifier results, a greater retrieval-practice effect on vividness was accompanied by a greater practice effect on reinstatement in the left PhC (i.e., reinstatement$_{recall}$ – reinstatement$_{baseline}$) (skipped Spearman's correlation: $r = .43$, 95% CI = [0.10 0.69]) but not in the right PhC (skipped Spearman's correlation: $r = .28$, 95% CI = [–0.09 0.59]) (*Appendix 1—figure 1d*).

**a** Retrieval practice preserves vividness

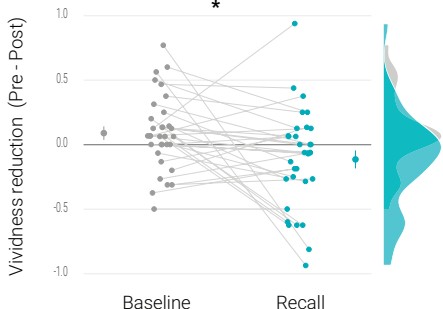

**b** Effects of retrieval practice on scene reactivation

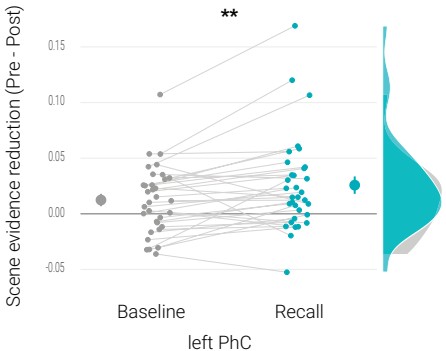

**c** Relationship between retrieval-practice effects on vividness and *scene reactivation.*

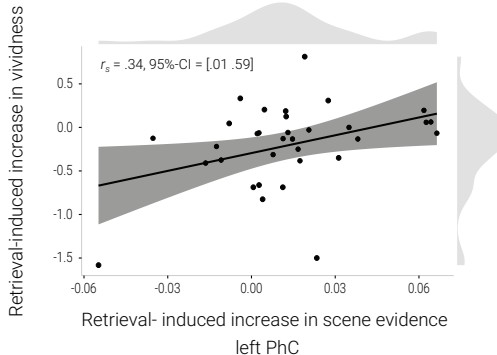

**d** Relationship between retrieval-practice effects on vividness and *reinstatement.*

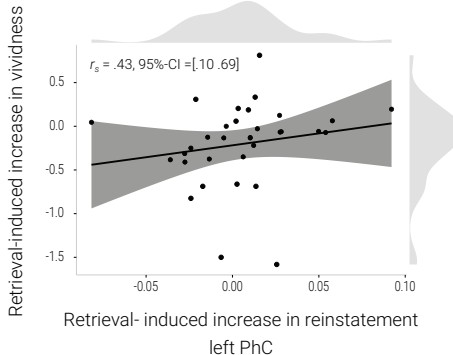

**Appendix 1—figure 1.** Effects of retrieval practice. (**a**) Retrieval retained vividness from the pre- to post-test as compared with the baseline condition. (**b**) Retrieval induced a reduction in scene evidence in left PhC: scene evidence decreased from the pre- to the post- test for recalled memories but not for baseline memories. Larger dots indicate the mean, error bars the standard error of the mean. (**c**) A greater above-baseline increase in vividness is associated with a greater above-baseline increase in scene evidence in left PhC as indicated by a

*Appendix 1—figure 1 continued on next page*

*Appendix 1—figure 1 continued*

robust skipped Spearman's correlation. (**d**) This greater above-baseline increase in vividness is also associated with a greater above-baseline increase in reinstatement in left PhC. Black lines indicate linear regression lines, dark grey shades indicate 95% - confidence intervals, PhC: parahippocampal cortex. ** *p* < .01, * *p* < .05, *n* = 33.

## Summary

The comparison of recall and baseline memories yielded the typical behavioral effect of retrieval practice (*Karpicke and Blunt, 2011*; *Karpicke and Roediger, 2008*; *Roediger and Butler, 2011*). In contrast, the analyses of the neural effects were somewhat inconclusive. Only the scene evidence obtained from the PhC showed a differential change for the practiced memories: compared to baseline memories, they exhibited reduced evidence for general scene reactivation. This is reminiscent of previous evidence that retrieval practice induces a greater change in parietal representations of memories that are later remembered than in those that are later forgotten (*Karlsson Wirebring et al., 2015*; *Liu et al., 2020*).

At the same time, a greater behavioral retrieval-practice effect was associated with both, a lesser reduction in scene reactivation and a more consistent reinstatement of individual memory representations. These associations suggest that retrieval-practice may facilitate the retention of a memory by stabilizing its neural representation (*Ferreira et al., 2019*).

Overall, the obtained pattern may indicate that retrieval practice can boost the retention of a memory in opposite fashion: on the one hand, by stabilizing its neural representation with high fidelity and, on the other hand, by changing the representation as a consequence of an ongoing semanticization (*Ferreira et al., 2019*) or in an attempt to reduce interference from competing memories (*Benoit et al., 2015*; *Kuhl et al., 2011*).

## Suppression-Induced Forgetting

This study set out to examine whether suppression affects the reactivation of avoided memory traces in a sustained fashion. Our central objective was the detection of scene reactivation and its subsequent modulation. We therefore adapted the Think/No-Think procedure by *Küpper et al., 2014* with the insertion of additional retrieval tests just before and after the suppression phase. This procedure further strengthens the memories (*Karpicke and Blunt, 2011*; *Karpicke and Roediger, 2008*; *Roediger and Butler, 2011*) compared with the original procedure and thus made it less likely that suppression would induce absolute forgetting rather than a gradual decline in the vividness of the memories (*Benoit et al., 2015*; *Ritvo et al., 2019*).

Nonetheless, as in the original procedure, participants in our study completed a final recall test. Here, they were presented with all of the objects and verbally described the respective associated scenes as detailed as possible within 15 s. The transcribed reports were then scored on *identification*, *details*, and *gist* (for details, see *Küpper et al., 2014*). On the *identification* measure, we scored a memory as correctly recalled if it included enough details so that the scene could unambiguously be identified. For the *detail* measure, we parsed the reports into smallest segments that independently conveyed information and counted the number of correct segments. The *gist* measure examined whether the reports entailed the pre-defined two to four elements that could not be changed or excluded without losing the main theme of the scene.

On all of these measures, recall performance was numerically worse for suppressed than for baseline memories (*Appendix 1—table 6*). However, as expected, with this modified procedure, suppression-induced forgetting (SIF) was not significant in the fMRI study (Gist: $t(32) = 0.33$, $p = .37$, $d = 0.06$, Details: $t(32) = 0.65$, $p = .26$, $d = 0.11$, Identification: $W = 85$, $p = .19$, $d = 0.17$). We obtained a similar pattern in the behavioral study with a significant effect for the identification measure only (Gist: $t(29) = 1.20$, $p = .12$, $d = 0.22$, Details: $t(29) = 0.88$, $p = .19$, $d = 0.16$, Identification: $W = 70.5$, $p = .038$, $d = 0.35$).

**Appendix 1—table 6.** Behavioral results of the final memory test.

| Study | Measure | Baseline M (SD) | Suppress M (SD) | Recall M (SD) |
|---|---|---|---|---|
| | Identification | 0.94 (0.18) | 0.92 (0.19) | 0.94 (0.18) |
| | Gist | 0.52 (0.20) | 0.48 (0.19) | 0.5 (0.19) |
| Behavioral Study | Details | 9.24 (2.71) | 9.07 (2.73) | 9.47 (2.84) |
| | Identification | 0.98 (0.05) | 0.96 (0.07) | 0.96 (0.06) |
| | Gist | 0.55 (0.15) | 0.54 (0.14) | 0.55 (0.15) |
| fMRI study | Details | 9.77 (1.61) | 9.66 (1.74) | 9.71 (1.75) |

However, we further scrutinized the evidence for SIF across the behavioral and fMRI study. We therefore computed a mini meta-analysis in R 4.0.3 (*R Development Core Team, 2020*) using the metafor 2.4–0 package (*Viechtbauer, 2010*). Specifically, we ran a three-level random effects model, given that the three outcome measures are not independent and nested within studies (*Cheung, 2014*). This model accounts for the variance in the observed effect sizes (level 1), variance between effect sizes within a study (level 2), and variance between studies (level 3).

Individual effect sizes for the comparison of baseline and suppress conditions were entered as standard mean change using raw score standardization. The model parameters were estimated using restricted maximum likelihood estimation (*Cheung, 2014*; *Viechtbauer, 2010*) with the Knapp & Hartung (2003) method for calculating regression coefficients and confidence intervals. Despite our modified procedure, this analysis did yield a, albeit non-significant, trend for a small SIF effect of 0.1, $SE$ = .04, $t$ = 2.49, $p$ = .055, 95% CI = [–0.003 0.203] with no significant heterogeneity between the studies $Q$ = 1.34, $p$ = .93.

Consistent with several reported dissociations in the extant literature (e.g. *Clark and Maguire, 2020*; *Levine et al., 2009*), the suppression reduced reductions in the objective memory measures were only partially associated with the reductions in the subjective measure of vividness. Only gist showed a significant skipped Spearman correlation in the MRI study ($r$ = 0.56, 95% CI = [0.29 0.78]) and a trend when pooling the data across the two studies ($r$ = 0.31, 95% CI = [–0.015 0.51]). There was no such effect when examining the behavioral study on its own ($r$ = –0.04, 95% CI = [–0.37 0.47]).

