## [Editor Report]

Meyer and Benoit explore the brain mechanisms underpinning memory suppression. The authors demonstrate that suppressed memories undergo sustained fading over time, which diminishes the vividness of the memory representation, and is associated with reduced reactivation of the memory representations in the parahippocampal cortex.

---

## [Decision Letter]

**Decision letter after peer review:**

Thank you for submitting your article "Suppression weakens unwanted memories via a sustained reduction of neural reactivation" for consideration by *eLife*. Your article has been reviewed by 3 peer reviewers, one of whom is a member of our Board of Reviewing Editors, and the evaluation has been overseen by Christian Büchel as the Senior Editor. The reviewers have opted to remain anonymous.

Essential revisions:

Methodology:

1. Inclusion/Exclusion criteria: There is no information on inclusion and exclusion criteria for the participants. The authors indicated that participants completed the Beck Depression Inventory, the Thought Control Ability Questionnaire and the State-Trait Anxiety Inventory. Were participants screened for the presence of psychiatric conditions and/or excluded based on BDI and STAI scores? Please, add any demographic information you may have on your participants (in the main text or on a Supplementary Table).

2. Procedure: Some information on the experimental procedure is not clearly represented in Figure 1A. A more detailed schematic description of the experimental procedure would be helpful including titles such as TNT phase, vividness test, etc.

3. The distinction between identification, gist and details measures of performance used in the final memory test should be briefly described (reference to Kupper et al., 2014 is not sufficient).

4. The aversive scenes used in the classifier training task are not sufficiently described. What are the 5 categories? How similar to the main task are the aversive scenes selected here?

5. TNT Phase: During the TNT phase, participant ratings of suppression success do not seem to have been acquired (e.g., intrusion rate). How did the authors ensure that the participants were actively trying to suppress or recall the aversive scenes during the TNT phase? Moreover, it cannot be determined whether consecutive suppression attempts are associated with a progressive decrease of intrusions over time.

6. Based on the TNT compliance questionnaire administered at the end of the experiment, please clarify which strategies the participants used (e.g., direct retrieval suppression, thought substitution) and what the exclusion criteria were based on this, along with reports of average compliance, and range.

7. The memory task involves aversive scenes from the IAPS that typically induce negative affect. Did the participants complete any questionnaire to rate the emotional valence and/or arousal of the aversive scenes before and after the TNT phase? Memory suppression may decrease the negative valence of the scenes.

Imaging Analysis:

8. ROI approach: Although there is a clear rationale for focusing on parahippocampal regions with regards to scene reactivation, it is not clear to me why a larger ROI was not used for the RSA analyses, especially since the RSA results were marginal. Perhaps the parahippocampal information is only part of what is being reconstituted for a specific memory, and as such there is not enough power to detect the reduction in strength of that memory. Perhaps effects would be found for visual areas and/or the amygdala. Visual areas seem like a potential candidate because other work has shown that visual areas are involved in memory suppression during early attempts at suppression and other works suggests that memory vividness is linked to processing in visual areas. The amygdala, which is associated with stimulus salience especially of an emotional nature, is another area that might be a good candidate ROI (or to be included in a large multi-region ROI).

9. Similarly, the linking of detail/vividness of memories to the PhC warrants further justification, particularly when other posterior parietal regions are also heavily implicated (e.g., angular gyrus, precuneus). Maybe I missed this, but I did not see any control regions included in the analysis to demonstrate the specificity of the PhC findings. To make firm conclusions about the right PhC could the authors test for the specificity between these associations (i.e., to show that right PhC is differentially implicated relative to the left). Figure 3 shows there is a lot of noise in the data.

10. Because this study demonstrates that memory suppression is linked to the classifier fit for scene retrieval it would be quite interesting to look at the degree to which connectivity between prefrontal cortex and parahippocampal regions are linked to memory suppression and vividness, that is, the data may be able to address the degree to which top-down mechanisms are driving the reduction in scene processing.

11. From Figure 1a, it appears that objects were always presented as the cue for subsequent recall or suppression. However, given that these objects were selected to be related to the aversive scenes, it is possible that the strong parahippocampal effects reported here reflect the scene-defining properties of these items. Work by Auger, Mullally and Maguire on scene defining objects seems particularly relevant in this context and warrants consideration.

12. Classifier results: Using the classifier, scene evidence is first compared between conditions globally across the grey matter. The results of this analysis are very global and do not highlight the specific regions involved. The decoding method seems to be able to distinguish brain states associated with the presentation of intact versus morphed aversive scenes. However, the classifier does not seem to provide any information on the neural substrate related to the task. For this reason, it does not provide any precise picture of the neural network involved. Why not select more than one region of interest (e.g., based on the TNT univariate analysis) which would have been more informative than looking at the whole grey matter? Is it possible to visualize the weighted maps obtained with the classifier (or the dot product of the weight patterns and t-maps)?

13. The authors used innovative multivariate pattern analyses such as machine learning and representational similarity analysis. However, the advantages of these methods in comparison to more traditional ones are not sufficiently highlighted. Please highlight the advantages of these methods to evidence the decreased neural reactivation of previously suppressed memories, in comparison to more classical methods (e.g., difference in activation at pre- vs post- tests).

14. Comparison of the differences in brain activity during the pre- and post- tests should be reported. This analysis should highlight the brain regions that are less activated following suppression.

15. Please also provide further information regarding the vividness data. Were there any relationships between vividness rate (or pre-post change in vividness) and the final recall? And brain activity during the TNT phase? Do individuals who better engage the memory control network during the TNT phase have less vivid memories for suppressed scenes?

Interpretation:

16. Could the authors comment on the role of individual differences in the capacity to suppress unwanted memories. The authors do attempt to explore individual differences in terms of the deterioration of vividness, however, there is quite a lot of variability in the data which suggests that other factors may be at play. For example, many of the figures suggest that the reported effects are not uniform – there seem to be a number of cases that decrease (i.e., show increased vividness) from baseline to suppression or remain flat/stable.

17. RSA results: In the Discussion (page 11, line 350), the authors state "…the absence of an overall effect might also reflect the varying degrees to which participants were successful at suppression. Indeed, right parahippocampal reinstatement was particularly affected in those people who also experienced the strongest decline in vividness." To test this hypothesis, it should be interesting to split the sample into groups with higher and lower memory control abilities based on either the Thought Control Ability Questionnaire or the vividness scales at post-test.

18. It appears that the results in the independent behavioural study did not fully replicate those of the neuroimaging study as the baseline memories show significant decline over time (in the fMRI study, baseline did not change significantly over time although p = .08 so maybe there is a trend there).

19. For the TNT univariate analysis, the MNI coordinates of the local maxima for the contrast Suppress < Recall are located mainly in posterior regions (cuneus, precuneus). How can the authors explain it? The medial posterior parietal cortex is considered as a memory engram (Brodt et al., 2018, Science). As such, the activity of this region may also be reduced during reactivation of suppressed items. Is the activity of the regions deactivated during the suppressed trials of the TNT related to the reduced global scene reactivation during suppression?

20. Authors should acknowledge the limitations of their study in the discussion.

*Reviewer #1 (Recommendations for the authors):*

I thoroughly enjoyed reading this manuscript. The experimental work and analyses are rigorously conducted and meticulously reported, and the manuscript itself is exceptionally well-written. I only have a few comments that I would like to see addressed.

1. Throughout the manuscript, I wondered about the role of individual differences in the capacity to suppress unwanted memories. Perhaps some individuals are naturally better at doing this than others, which in turn might confer resilience in the face of trauma or conversely an impaired capacity to suppress memories might confer risk for PTSD. The authors do attempt to explore individual differences in terms of the deterioration of vividness, however, there is quite a lot of variability in the data which suggests that other factors may be at play. For example, many of the figures suggest that the reported effects are not uniform - there seem to be a number of cases that decrease (i.e., show increased vividness) from baseline to suppression or remain flat/stable. I would have liked to have seen these issues explored in more depth. Do the authors have any ratings of task difficulty or individual differences in imagery capacity that might speak to this issue?

2. I would have liked to see stronger justification for the a priori focus on the PhC, given that both the PhC and hippocampus demonstrated significant deactivation during suppression trials and the well-established contribution of the hippocampus in scene cognition. Similarly, the linking of detail/vividness of memories to the PhC warrants further justification, particularly when other posterior parietal regions are also heavily implicated (e.g., angular gyrus, precuneus). Maybe I missed this, but I did not see any control regions included in the analysis to demonstrate the specificity of the PhC findings.

3. While I appreciate that the focus of this study is on the PhC, I wondered about the finding of increased dlPFC activation during memory suppression and how this might relate to individual differences in cognitive control? This likely lies outside the scope of the current study but I would be keen to hear the authors' thoughts.

4. From Figure 1a, it appears that objects were always presented as the cue for subsequent recall or suppression. However, given that these objects were selected to be related to the aversive scenes, it is possible that the strong parahippocampal effects reported here reflect the scene-defining properties of these items. Work by Auger, Mullally and Maguire on scene defining objects seems particularly relevant in this context and warrants consideration.

5. It appears that the results in the independent behavioural study did not fully replicate those of the neuroimaging study as the baseline memories show significant decline over time (in the fMRI study, baseline did not change significantly over time although p = .08 so maybe there is a trend there).

6. To make firm conclusions about the right PhC could the authors test for the specificity between these associations (i.e., to show that right PhC is differentially implicated relative to the left). Figure 3 shows there is a lot of noise in the data.

*Reviewer #2 (Recommendations for the authors):*

Although the authors have performed a logical and careful set of analyses, there is likely more that could be leveraged with their data set. Some suggestions are provided below.

First, although there is a clear rationale for focusing on parahippocampal regions with regards to scene reactivation, it is not clear to me why a larger ROI was not used especially for the RSA analyses, especially since the RSA results were marginal. Perhaps the parahippocampal information is only part of what is being reconstituted for a specific memory, and as such there is not enough power to detect the reduction in strength of that memory. Perhaps effects would be found for visual areas and/or the amygdala. Visual areas seem like a potential candidate because other work has shown that visual areas are involved in memory suppression during early attempts at suppression and other works suggests that memory vividness islinked to processing in visual areas. The amygdala, which is associated with stimulus salience especially of an emotional nature, is another area that might be a good candidate ROI (or to be included in a large multi-region ROI).

Second, because this study demonstrates that memory suppression is linked to the classifier fit for scene retrieval it would be quite interesting to look at the degree to which connectivity between prefrontal cortex and parahippocampal regions are linked to memory suppression and vividness, that is, the data may be able to address the degree to which top-down mechanisms are driving the reduction in scene processing.

*Reviewer #3 (Recommendations for the authors):*

General comment: This paper deals with an interesting question, the hypotheses are clearly stated, and it uses cutting-edge neuroimaging methods to answer the main hypotheses. The study is carefully conducted. However, I have some comments, which need to be addressed before publication (see public review and below).

– Participants inclusion: There is no information on inclusion and exclusion criteria for the participants. However, it is important to control for the presence of psychiatric comorbidities. Indeed, there is evidence that individuals with psychological disorders (e.g., depression, PTSD) are less efficient to control their memories and exhibit different neural patterns during the Think/No-Think task in comparison to healthy controls. The authors indicated that participants completed the Beck's Depression Inventory, the Thought Control Ability Questionnaire and the State-Trait Anxiety Inventory. Were participants screened for the presence of psychiatric conditions and/or excluded based on their scores at the Beck and STAI? If not, please comment on it. Can the authors share any other demographic characteristics of their sample? Please, add any demographic information you may have on your participants (in the main text or on a Supplementary Table).

– Procedure and material (clarification):

– Some information on the experimental procedure is not clearly represented in Figure 1A. I suggest to provide a more detailed schematic description of the experimental procedure. For example, titles such as TNT phase, vividness test, etc. should be added to make the whole procedure easier to catch.

– The distinction between identification, gist and details measures of performance used in the final memory test should be shortly described, reference to Kupper et al., (2014) is not sufficient.

– The aversive scenes used in the classifier training task are not sufficiently described. What are the 5 categories? How similar to the main task are the aversive scenes selected here?

– Information in Figure 1C and Appendix 2 seems to indicate that there is a group of participants that was tested only behaviorally (“behavioral study”). I did not see any mention of this behavioral study in the methods or there is something unclear that I misunderstood. Please clarify what is the additional value of this behavioral study.

– Multivariate methods: Authors used innovative multivariate pattern analyses such as machine learning and representational similarity analysis. However, the advantages of these methods in comparison to more traditional ones are not sufficiently highlighted. Please highlight the advantages of these methods to evidence the decreased neural reactivation of previously suppressed memories, in comparison to more classical methods (e.g., difference in activation at pre- vs post- tests).

– TNT analyses:

– For the TNT univariate analysis, the MNI coordinates of the local maxima for the contrast Suppress < Recall are located mainly in posterior regions (cuneus, precuneus). How can the authors explain it? The medial posterior parietal cortex is considered as a memory engram (Brodt et al., 2018, Science). As such, the activity of this region may also be reduced during reactivation of suppressed items.

– Is the activity of the regions deactivated during the suppressed trials of the TNT related to the reduced global scene reactivation during suppression?

– Pre- and post- test analyses

– Comparison of the differences in brain activity during the pre- and post- tests should be reported. This analysis should highlight the brain regions that are less activated following suppression.

– Please report more on the vividness data. Were there any relationships between vividness rate (or pre-post change in vividness) and the final recall? And brain activity during the TNT phase? We can assume that individuals who engage more the memory control network during the TNT phase have less vivid memories for suppressed scenes.

– Classifier results:

– During the classifier training task, participants had to perform a one-back task to ensure they paid attention to the task. How was the performance of the participants?

– Using the classifier, scene evidence is first compared between conditions globally across the grey matter. The results of this analysis are very global and do not highlight the specific regions involved. The decoding method seems to be able to distinguish brain states associated with the presentation of intact versus morphed aversive scenes. However, the classifier does not seem to provide any information on the neural substrate related to the task. For this reason, it does not provide any precise picture of the neural network involved. Why not selecting more than one region of interest (e.g., based on the TNT univariate analysis) which would have been more informative than looking at the whole grey matter? Is it possible to visualize the weight maps obtained with the classifier (or the dot product of the weight patterns and t-maps)? Does it make any sense?

– RSA results: Authors wrote (discussion, page 11, line 350): “…the absence of an overall effect might also reflect the varying degrees to which participants were successful at suppression. Indeed, right parahippocampal reinstatement was particularly affected in those people who also experienced the strongest decline in vividness.” To test this hypothesis, it should be interesting to split the sample into groups with higher and lower memory control abilities based on either the Thought Control Ability Questionnaire or the vividness scales at post-test.

– Behavioral results:

– During the TNT phase, participants were not asked to rate their capacity to succeed when attempted to suppress or retrieve the aversive scenes (e.g., intrusion rate). How can the authors ensure that the participants were actively trying to suppress or recall the aversive scenes during the whole TNT phase? Moreover, it cannot be determined whether consecutive suppression attempts are associated with a progressive decrease of intrusions over time.

– Based on the TNT compliance questionnaire administered at the end of the experiment, please clarify which strategies the participants used (e.g., direct retrieval suppression, thought substitution) and what the exclusion criteria were based on this, along with reports of average compliance, and range.

– The memory task involves aversive scenes from the IAPS that typically induce negative affects. Did the participants complete any questionnaire to rate the emotional valence and/or arousal of the aversive scenes before and after the TNT phase? Memory suppression may decrease the negative valence of the scenes.

– Statistical tests showed a reduction in vividness for suppressed, but also for baseline memories. Please, comment on this effect for baseline memories.

– Authors should acknowledge the limitations of their study in the discussion.

---

## [Author Response]

Essential revisions:Methodology:1. Inclusion/Exclusion criteria: There is no information on inclusion and exclusion criteria for the participants. The authors indicated that participants completed the Beck Depression Inventory, the Thought Control Ability Questionnaire and the State-Trait Anxiety Inventory. Were participants screened for the presence of psychiatric conditions and/or excluded based on BDI and STAI scores? Please, add any demographic information you may have on your participants (in the main text or on a Supplementary Table).

We had screened our participants for the absence of self-reported psychological or neurological disorders before the experiment. No participants were subsequently excluded based on the BDI or STAI. We included these questionnaires to allow for a demographic description of our sample and to possibly explore relationships between these measures and forgetting (see also essential comment #17). As mentioned in the manuscript and further discussed below (essential comment #7), we did assess non-compliance with the task instructions on a post-experimental questionnaire. Based on standard procedures (Hertel and Calcaterra, 2005; Liu et al., 2021), this questionnaire led to the exclusion of one participant in the MRI and two participants in the behavioral study.

We provide more details about our exclusion criteria in the methods (p. 28, l. 671ff, p.34, l. 781ff) and have also added a table including demographic information and the questionnaire scores to the supplement (supplementary table 1).

2. Procedure: Some information on the experimental procedure is not clearly represented in Figure 1A. A more detailed schematic description of the experimental procedure would be helpful including titles such as TNT phase, vividness test, etc.

We thank the reviewers for suggesting improvements to the figure. In the revised version, we have adopted the suggested titles and added further information about the timings of individual trials. We have also highlighted the parts of the study that were conducted in the MRI scanner. Moreover, we have provided further procedural details in the figure caption and refer to the methods for a comprehensive description.

3. The distinction between identification, gist and details measures of performance used in the final memory test should be briefly described (reference to Kupper et al., 2014 is not sufficient).

We agree that a description is warranted to make this part of the manuscript self-contained.

We added a brief description of those measures to the supplement (p.10 , l. 163 ff).

4. The aversive scenes used in the classifier training task are not sufficiently described. What are the 5 categories? How similar to the main task are the aversive scenes selected here?

The critical categories were aversive scenes and their morphed versions. The aversive scenes were taken from the same (i.e., IAPS) and similar (i.e., EmoPic (Wessa et al., 2010)), GAPED (Dan-Glauser and Scherer, 2011), NAPS (Marchewka et al., 2014) databases as the pictures presented in the main task. They were all categorized as *aversive* based on their normed valence and/or arousal ratings. The pictures were chosen to also resemble the ones used for the original task in terms of complexity, closeness, and the degree to which they include humans. We moreover ensured that they depict aversive themes that also featured in the task proper (i.e., fight, accident, disaster, war, hospital, injury). As additional filler categories, the classifier training task also presented pictures of neutral scenes, fruits, and objects.

We have provided the additional details about the classifier training task in the revised methods section (p.30, l.693 ff).

5. TNT Phase: During the TNT phase, participant ratings of suppression success do not seem to have been acquired (e.g., intrusion rate). How did the authors ensure that the participants were actively trying to suppress or recall the aversive scenes during the TNT phase? Moreover, it cannot be determined whether consecutive suppression attempts are associated with a progressive decrease of intrusions over time.

It is correct that we did not acquire intrusion ratings and thus have no behavioral measure of a change in intrusiveness over consecutive suppression attempts. However, in this particular study, we were primarily interested in the sustained after-effects of suppression rather than the gradual changes over the course of the Think/No-Think phase.

Nonetheless, we have a number of good reasons to believe that participants indeed performed the task as instructed. First, the below-baseline reduction in vividness indicates that the participants engaged a mechanism that detrimentally affected the memories. Notably, we observed the opposite pattern of above-baseline enhancement in vividness for the recall condition.

Second, the activity during the suppression phase showed clear differences for recall versus suppress trials. This pattern closely resembles previous findings in the extant literature, including those derived from studies that had used intrusion ratings as an online measure of task performance (e.g. Benoit et al., 2015; Levy and Anderson, 2012; Mary et al., 2020). In addition, our pattern classifier also yielded reduced scene evidence during the suppression than during the recall trials. This finding further corroborates that participants were successful at performing the task as instructed.

Finally, we assessed overall compliance via a post-experimental questionnaire. Based on this questionnaire, we excluded the one participant of the MRI study and two participants of the behavioral study, who admitted to not have performed the task as instructed (please also see our response to essential comment #7). Moreover, on the postexperimental questionnaire, participants indicated that they successfully suppressed the scenes on about 75% of the suppress trials (scale from zero (0%) to four (100%); M = 2.99, SD = 0.92) and that they remembered the respective scenes on about 86% (M = 3.44, SD = 0.88) of the recall trials.

In the revised manuscript, we have included the additional evidence corroborating that participants generally complied with the task instructions and that they were successful at employing retrieval suppression as a means of avoiding the unwanted memories (p. 34, l.792ff). However, when discussing the limitations of our study, we now also note the potential of including intrusion ratings as an online measure of suppression success (p.27, l.614ff).

6. Based on the TNT compliance questionnaire administered at the end of the experiment, please clarify which strategies the participants used (e.g., direct retrieval suppression, thought substitution) and what the exclusion criteria were based on this, along with reports of average compliance, and range.

Our compliance questionnaire was designed to assess whether participants intentionally tried to recall the memories of the suppress condition and thus disregarded the task instructions. Specifically, they answered three questions: On a scale from zero (never) to four (very often), participants rated if they had *intentionally* recalled the suppress items, either (i) during the presentation of the reminders or (ii) following their offset or (iii) if they had generally made an effort to recall scenes of the suppress condition. The sum of the three scores serves as a measure of task compliance. The mean compliance score in the final MRI sample was 0.67 (SD = 0.96, range: 0-3). Following standard procedure (van Schie et al., 2013 see also Liu et al., 2021), this sample excluded the one participant with a score of four. In the behavioral study, non-compliance led to the exclusion of two participants. On the post-experimental questionnaire, participants further indicated that they had engaged more frequently in direct suppression than in the alternate strategy of thought substitution (scale from zero (never) to four (always), mean difference: 2.12, t(32)=9.8 , p <.001).

We have included more details about the compliance questionnaire in the methods section and also report the adherence to direct suppression rather than thought substitution (p.34, l.792ff).

7. The memory task involves aversive scenes from the IAPS that typically induce negative affect. Did the participants complete any questionnaire to rate the emotional valence and/or arousal of the aversive scenes before and after the TNT phase? Memory suppression may decrease the negative valence of the scenes.

We agree that this is an important research question. In the current study, we did not assess ratings of negative affect and thus can’t speak to a possible change in subjective emotional experience. In the revised manuscript, we discuss this as a limitation of the study (p.27 l. 623ff). However, we also refer to the work by Gagnepain et al., (2017), who did show that suppression can affect ratings of valence.

Imaging Analysis:8. ROI approach: Although there is a clear rationale for focusing on parahippocampal regions with regards to scene reactivation, it is not clear to me why a larger ROI was not used for the RSA analyses, especially since the RSA results were marginal. Perhaps the parahippocampal information is only part of what is being reconstituted for a specific memory, and as such there is not enough power to detect the reduction in strength of that memory. Perhaps effects would be found for visual areas and/or the amygdala. Visual areas seem like a potential candidate because other work has shown that visual areas are involved in memory suppression during early attempts at suppression and other works suggests that memory vividness is linked to processing in visual areas. The amygdala, which is associated with stimulus salience especially of an emotional nature, is another area that might be a good candidate ROI (or to be included in a large multi-region ROI).

We had focused our analysis on the PhC because we had considered this region to be most sensitive to a sustained disruption of neural reinstatement. However, we did not mean to suggest that it is the only region that may show such an effect.

Following this comment as well as the related essential comment #10, we have explored the activity patterns in bilateral masks of V1, amygdala, precuneus, and angular gyrus. All of these regions, apart from the amygdala, show greater same- than different-item similarity. However, unlike the PhC, none of them exhibited a correlation between reduced reinstatement and reduced vividness. They also showed no evidence for a suppression-induced reduction in reinstatement. For these candidate control regions, we thus have no additional data to corroborate that the RSA effect reflects replicable reinstatement of memory representations rather than the repeated presentation of the same reminder objects. In a further exploratory analysis, we combined all of these regions in a multi-region ROI. Again, we did not see any evidence for either a suppression-induced reduction in reinstatement or a correlation between reduced reinstatement and reduced vividness.

A similar picture emerged when we explored these additional regions using a pattern classifier. We successfully classified patterns associated with aversive versus morphed scenes in each of these regions. Of note, though, in each case, the classification was numerically worse than the classification based on the data from the right PhC – an observation that further corroborates the preferential involvement of this region in the processing of scenes. However, none of the additional regions showed a significant suppression-induced reduction in scene evidence, and only for the angular gyrus and precuneus did we observe a trend for a correlation with suppression-induced reductions in vividness.

We have included the additional analyses in supplementary table 5 and summarize them in the result section c In general, the obtained pattern further highlights that the PhC may be particularly sensitive to changes in the representations for memories for scenes. However, in the revised discussion, we also note that representations in other regions may be more sensitive to other qualities of a memory. Specifically, as suggested by the reviewer, the amygdala may be more sensitive to suppression-induced changes in subjective emotional experience ( p.27, l.623ff).

9. Similarly, the linking of detail/vividness of memories to the PhC warrants further justification, particularly when other posterior parietal regions are also heavily implicated (e.g., angular gyrus, precuneus). Maybe I missed this, but I did not see any control regions included in the analysis to demonstrate the specificity of the PhC findings. To make firm conclusions about the right PhC could the authors test for the specificity between these associations (i.e., to show that right PhC is differentially implicated relative to the left). Figure 3 shows there is a lot of noise in the data.

Indeed, we did not mean to suggest that suppression-induced reductions in vividness are solely attributable to reductions in PhC reactivation. We had focused on this region because of its preferential role in reinstating scenes (Staresina et al., 2011) and its documented association with vividness of these memories (Kensinger et al., 2011; Sheldon and Levine, 2013; Todd et al., 2013) We had thus reasoned that PhC reactivation should be a particularly sensitive marker of vividness and its sustained modulation by suppression.

Given the proposed contribution of the angular gyrus and the precuneus to the phenomenological experience of a memory (e.g., Richter, Cooper, et al., 2016), we were of course happy to further examine the pattern of reactivation in these regions. In response to essential comment #9, we also examined such possible effects in V1 and the amygdala. However, of these candidate control regions, only the angular gyrus and precuneus showed a trend for an association between suppression-induced changes in scene evidence and in vividness.

As for the PhC effects, we are reluctant to draw any conclusions about hemispheric differences. Though both the RSA analysis and the pattern classifier yielded significant relationships with the reduction in vividness in right PhC only, at least the classifier data also showed a trend for the left PhC. Finally, we note that our inferences are based on robust skipped spearman correlations that are better suited for noisy data given that they are less susceptible to outliers.

In the revised manuscript, we have taken several actions to address this comment. First, in the introduction (p.5, l.103ff) and in the methods (p. 37 l.856ff), we more carefully describe that we focus on the PhC because of its presumed sensitivity for changes in scene reactivation and accompanying reductions in vividness. We also indicate that we don’t expect this effect to solely rely on the PhC. Second, we report the results of the additional candidate control regions in supplementary table 5 and summarize them in the result section (p.16, l.358ff, p.19 l.433ff). Finally, we ensured that we don’t assign a particular significance to the apparent lateralization, while being clear that we only obtained significant correlations for the data from the right hemisphere (p. 23 l. 519ff).

10. Because this study demonstrates that memory suppression is linked to the classifier fit for scene retrieval it would be quite interesting to look at the degree to which connectivity between prefrontal cortex and parahippocampal regions are linked to memory suppression and vividness, that is, the data may be able to address the degree to which top-down mechanisms are driving the reduction in scene processing.

We agree that this is an interesting question. We took this comment to run an exploratory PPI analysis (e.g., Paz-Alonso et al., 2013). It was seeded at the peak activation in the right dlPFC and set up to reveal increases in connectivity during suppression versus recall trials. We could thus explore the associations between the coupling parameters in the PhC and markers of suppression success. However, we did not find evidence for a correlation with either the reduction in vividness, scene evidence, or reinstatement. Given the inconclusiveness of these exploratory analyses, we opt to not report them in the manuscript.

11. From Figure 1a, it appears that objects were always presented as the cue for subsequent recall or suppression. However, given that these objects were selected to be related to the aversive scenes, it is possible that the strong parahippocampal effects reported here reflect the scene-defining properties of these items. Work by Auger, Mullally and Maguire on scene defining objects seems particularly relevant in this context and warrants consideration.

We thank the reviewer for pointing us towards the paper by Auger et al., 2012, which we have read with great interest. The authors observe greater univariate activity in the PhC when participants view objects that they had deemed as particular space defining (e.g., a lighthouse is more space defining than a car). They thus conclude that this region may have a particular role in the construction of spatial representations. However, it seems plausible that this effect may also reflect the retrieval of particular scenes that is triggered by the presentation of the objects.

We also note that the objects in our procedure were chosen precisely because they were merely peripheral to the scene (see also response to essential comment #5). That is, they were explicitly chosen because they were *not* space defining for our particular scenes. Moreover, we counterbalanced the assignment of the object-scene pairs to the three conditions (baseline, suppress, recall). Incidental differences in space-defining properties of the objects could thus not account for the observed pattern.

With reference to Auger et al., we now more strongly stress the nature of the chosen objects and their relationship to the paired scenes (p.30, l.677ff).

12. Classifier results: Using the classifier, scene evidence is first compared between conditions globally across the grey matter. The results of this analysis are very global and do not highlight the specific regions involved. The decoding method seems to be able to distinguish brain states associated with the presentation of intact versus morphed aversive scenes. However, the classifier does not seem to provide any information on the neural substrate related to the task. For this reason, it does not provide any precise picture of the neural network involved. Why not select more than one region of interest (e.g., based on the TNT univariate analysis) which would have been more informative than looking at the whole grey matter? Is it possible to visualize the weighted maps obtained with the classifier (or the dot product of the weight patterns and t-maps)?

To answer our research question, we took two complementary approaches. On the one hand, we took a global view and examined reactivation across the whole greymatter. On the other hand, we zoomed in on the PhC as a region that should be particularly sensitive to sustained disruptions in memory reactivation. This region also showed reduced activity during the suppression phase. Indeed, we have now conducted additional exploratory analyses of further ROIs, also in response to comments #9 and #10. These further suggest a particular role for the PhC. However, our analyses also indicate that the effects across the greymatter are not solely driven by the PhC, indicating that it is unlikely to be the only region contributing to this effect.

In addition to reporting further exploratory analyses of other candidate control regions, we have taken the comment to upload the average weight pattern to neurovault (https://neurovault.org/collections/KAZGAACE/). This map shows which regions particularly contributed to the classifier, and thus collectively also to the changes in scene evidence.

13. The authors used innovative multivariate pattern analyses such as machine learning and representational similarity analysis. However, the advantages of these methods in comparison to more traditional ones are not sufficiently highlighted. Please highlight the advantages of these methods to evidence the decreased neural reactivation of previously suppressed memories, in comparison to more classical methods (e.g., difference in activation at pre- vs post- tests).

In general, MVPA methods often have a greater sensitivity to experimental manipulations than univariate methods (Norman et al., 2006). That is, the former can also exploit subtle differences in activity patterns across voxels, whereas the latter are confined to uniform BOLD signal differences between conditions. Indeed, only a few clusters across the brain exhibited a suppression-induced reduction in univariate brain activity (see response to essential comment #15). Critically, the multivariate approaches moreover allow us to identify changes in activity patterns that are more *specific* to the processing of scene information. RSA has the further benefit that it allows for investigating the reinstatement of individual memory representations.

In the revised manuscript, we report the univariate results and take them as an opportunity to highlight the benefits of the employed MVPA techniques (p.9, l.206ff).

14. Comparison of the differences in brain activity during the pre- and post- tests should be reported. This analysis should highlight the brain regions that are less activated following suppression.

In the revised manuscript, we follow the suggestion to include a new analysis of the univariate pre- to post change induced by suppression, i.e., the contrast (post: baseline>suppress) > (pre: baseline > suppress). This analysis yielded a few clusters, which we report in the Results section (p.9, l.199ff) and supplementary table 4.

15. Please also provide further information regarding the vividness data. Were there any relationships between vividness rate (or pre-post change in vividness) and the final recall? And brain activity during the TNT phase? Do individuals who better engage the memory control network during the TNT phase have less vivid memories for suppressed scenes?

We explored the association between vividness change and suppression-induced forgetting (SIF) on the final recall test. We did not find a correlation of vividness reduction with SIF on either the details or the identification measure. However, we did observe a significant correlation with the gist measure in the MRI study (r = 0.56, CI = [0.29 0.78]). There was also a trend for this effect when pooling the data across the two studies (r = 0.31 , CI = [-0.02 0.51]).

The objective and subjective forgetting measures were thus only partially associated with each other, consistent with several reported dissociations in the extant literature (e.g., Clark and Maguire, 2020; Levine et al., 2009). We have included the relationship between the reductions in vividness and in performance on the final recall task in the supplement (p.12, ll.195ff)

In response to the latter part of this comment, we also examined the association between suppression-related activation of the right dlPFC (as the central region of the memory control network) and the ensuing suppression-induced reduction in vividness. However, the correlation between these measures was not significant (r = -0.25, CI = [-0.55 0.2]). See also comment #11 for the association between dlPFC-PhC connectivity and vividness change.

Interpretation:16. Could the authors comment on the role of individual differences in the capacity to suppress unwanted memories. The authors do attempt to explore individual differences in terms of the deterioration of vividness, however, there is quite a lot of variability in the data which suggests that other factors may be at play. For example, many of the figures suggest that the reported effects are not uniform – there seem to be a number of cases that decrease (i.e., show increased vividness) from baseline to suppression or remain flat/stable.

We agree that understanding individual differences is a critical goal for research on memory suppression. This is particularly the case given the clinical relevance due to its association with PTSD, anxiety, and depression (Catarino et al., 2015; Mary et al., 2020; Stramaccia et al., 2020). In our study, participants completed the *Thought Control Ability Questionnaire* (TCAQ), the *Beck Depression Inventory*, and the *State-Trait Anxiety Inventory*. These inventories help us define the demographics of our sample. They also allow us to examine relationships with the suppression-induced reduction in vividness. However, only the TCAQ showed a trend for a correlation in the MRI study (r = 0.32 , CI = [-0.08 0.58]) and a significant correlation across the two studies (r = 0.27, CI = [0.004 0.0.49]) (see Küpper et al., 2014). There was thus just some evidence for an association between individual differences in self-reported control ability and in the impact of suppression.

We report these exploratory correlations in the revised supplement (supplementary table 1).

17. RSA results: In the Discussion (page 11, line 350), the authors state "…the absence of an overall effect might also reflect the varying degrees to which participants were successful at suppression. Indeed, right parahippocampal reinstatement was particularly affected in those people who also experienced the strongest decline in vividness." To test this hypothesis, it should be interesting to split the sample into groups with higher and lower memory control abilities based on either the Thought Control Ability Questionnaire or the vividness scales at post-test.

Yes, we agree that this is an important point – though we suggest that it is better addressed with a correlation rather than a median split. Indeed, consistent with the hypothesis, we had observed a positive correlation between suppression-induced reductions in vividness and suppression-induced reductions in both general scene evidence and memory-specific reinstatement. However, we have now also looked into the relationship between the TCAQ and the two neural measures. The correlations were not significant.

18. It appears that the results in the independent behavioural study did not fully replicate those of the neuroimaging study as the baseline memories show significant decline over time (in the fMRI study, baseline did not change significantly over time although p = .08 so maybe there is a trend there).

We agree that the two studies differ with respect to the rate of passive forgetting simply occurring with the passage of time. That is, we observed such passive forgetting for the baseline memories in the MRI study only. However, we note that we measure these baseline effects precisely to assess whether suppression yields forgetting that exceeds the respective rate of passive effects occurring in a given study.

We have clarified that the studies yield the same pattern with regards to suppression-induced changes, though they somewhat differ with respect to the changes in baseline vividness (p.8, l.187ff).

19. For the TNT univariate analysis, the MNI coordinates of the local maxima for the contrast Suppress < Recall are located mainly in posterior regions (cuneus, precuneus). How can the authors explain it? The medial posterior parietal cortex is considered as a memory engram (Brodt et al., 2018, Science). As such, the activity of this region may also be reduced during reactivation of suppressed items. Is the activity of the regions deactivated during the suppressed trials of the TNT related to the reduced global scene reactivation during suppression?

Indeed, consistent with prior studies and as shown in supplementary table 2, there are quite a number of regions that exhibit reduced activation during suppression. These regions differ with respect to the *t*-values at their respective peaks. We do note though that it is difficult to compare *t*-values across different brain regions. For example, a greater effect in posterior cortical regions may reflect regional differences in SNR rather than stronger task-related activation. However, as discussed in essential comments #9 and #10, we have now also examined memory reactivation in the precuneus as an additional candidate control region.

Moreover, following the latter part of this comment, we have examined the relationship between suppression-related deactivations during the suppression phase and suppression-induced reductions in reactivation. We performed these analyses separately across the greymatter mask, the precuneus, and the PhC. However, none of the analyses yielded significant correlations.

As discussed above, we have included the results from the precuneus in the revised manuscript. We also report that there was no clear evidence of a direct relationship between deactivation during suppression and the sustained reductions in reactivation in the PhC (p.16, l. 352ff).

20. Authors should acknowledge the limitations of their study in the discussion.

In the revised manuscript, we have made sure to more explicitly describe the limitations of this study in a new section of the discussion (p.27, l.614). In particular, this section refers to the possible benefits of including intrusion ratings in future studies, the question about the attenuation of emotions, and the causal relationship between the neural suppression mechanism and the sustained reduction in neural reactivation.

Reviewer #3 (Recommendations for the authors):– Procedure and material (clarification):– Information in Figure 1C and Appendix 2 seems to indicate that there is a group of participants that was tested only behaviorally ("behavioral study"). I did not see any mention of this behavioral study in the methods or there is something unclear that I misunderstood. Please clarify what is the additional value of this behavioral study.

Given the novelty of the modified pre-post design and of the suppression-induced reduction in vividness, we wanted to ensure the robustness of this effect. We thus ran a behavioral study with an independent sample.

We describe the rationale for the study as well as the study characteristics in the revised methods section (p.29, l.659 ff) and in supplementary table 1.

– Classifier results:– During the classifier training task, participants had to perform a one-back task to ensure they paid attention to the task. How was the performance of the participants?

As expected, participants performed very well on the one-back task (% correct: m = 97.6% of trials (SD = 2.83)).

We have described the performance on the one-back task in the revised methods section (p.33 l.774f).

References

Anderson, M. C., Ochsner, K. N., Kuhl, B., Cooper, J., Robertson, E., Gabrieli, S. W., Glover, G. H., and Gabrieli, J. D. E. (2004). Neural systems underlying the suppression of unwanted memories. *Science (New York, N.Y.)*, *303*(5655), 232–235. https://doi.org/10.1126/science.1089504

Auger, S. D., Mullally, S. L., and Maguire, E. A. (2012). Retrosplenial Cortex Codes for Permanent Landmarks. *PLOS ONE*, *7*(8), e43620. https://doi.org/10.1371/journal.pone.0043620

Benoit, R. G., and Anderson, M. C. (2012). Opposing Mechanisms Support the Voluntary Forgetting of Unwanted Memories. *Neuron*, *76*(2), 450–460. https://doi.org/10.1016/j.neuron.2012.07.025

Benoit, R. G., Hulbert, J. C., Huddleston, E., and Anderson, M. C. (2015). Adaptive Top–Down Suppression of Hippocampal Activity and the Purging of Intrusive Memories from Consciousness. *Journal of Cognitive Neuroscience*, *27*(1), 96–111. https://doi.org/10.1162/jocn_a_00696

Catarino, A., Kupper, C. S., Werner-Seidler, A., Dalgleish, T., and Anderson, M. C. (2015). Failing to Forget: Inhibitory-Control Deficits Compromise Memory Suppression in Posttraumatic Stress Disorder. *Psychological Science*, *26*(5), 604–616. https://doi.org/10.1177/0956797615569889

Clark, I. A., and Maguire, E. A. (2020). Do questionnaires reflect their purported cognitive functions? *Cognition*, *195*, 104114. https://doi.org/10.1016/j.cognition.2019.104114

Dan-Glauser, E. S., and Scherer, K. R. (2011). The Geneva Affective Picture Database (GAPED): A new 730-picture database focusing on valence and normative significance. *Behavior Research Methods*, *43*(2), 468–477. https://doi.org/10.3758/s13428-011-0064-1

Depue, B. E., Burgess, G. C., Willcutt, E. G., Ruzic, L., and Banich, M. T. (2010). Inhibitory control of memory retrieval and motor processing associated with the right lateral prefrontal cortex: Evidence from deficits in individuals with ADHD. *Neuropsychologia*, *48*(13), 3909–3917. https://doi.org/10.1016/j.neuropsychologia.2010.09.013

Depue, B. E., Curran, T., and Banich, M. T. (2007). Prefrontal Regions Orchestrate Suppression of Emotional Memories via a Two-Phase Process. *Science*, *317*(5835), 215–219. https://doi.org/10.1126/science.1139560

Ehlers, A. (2010). Understanding and Treating Unwanted Trauma Memories in Posttraumatic Stress Disorder. *Zeitschrift Fur Psychologie*, *218*(2), 141–145. https://doi.org/10.1027/0044-3409/a000021

Ehlers, A., and Clark, D. M. (2000). A cognitive model of posttraumatic stress disorder. *Behaviour Research and Therapy*, *38*(4), 319–345. https://doi.org/10.1016/s0005-7967(99)00123-0

Gagnepain, P., Hulbert, J., and Anderson, M. C. (2017). Parallel Regulation of Memory and Emotion Supports the Suppression of Intrusive Memories. *The Journal of Neuroscience*, *37*(27), 6423–6441. https://doi.org/10.1523/JNEUROSCI.2732-16.2017

Hertel, P. T., and Calcaterra, G. (2005). Intentional forgetting benefits from thought substitution. *Psychonomic Bulletin and Review*, *12*(3), 484–489. https://doi.org/10.3758/BF03193792

Kensinger, E. A., Addis, D. R., and Atapattu, R. K. (2011). Amygdala activity at encoding corresponds with memory vividness and with memory for select episodic details. *Neuropsychologia*, *49*(4), 663–673. https://doi.org/10.1016/j.neuropsychologia.2011.01.017

Küpper, C. S., Benoit, R. G., Dalgleish, T., and Anderson, M. C. (2014). Direct suppression as a mechanism for controlling unpleasant memories in daily life. *Journal of Experimental Psychology: General*, *143*(4), 1443–1449. https://doi.org/10.1037/a0036518

Levine, B., Svoboda, E., Turner, G. R., Mandic, M., and Mackey, A. (2009). Behavioral and functional neuroanatomical correlates of anterograde autobiographical memory in isolated retrograde amnesic patient M.L. *Neuropsychologia*, *47*(11), 2188–2196. https://doi.org/10.1016/j.neuropsychologia.2008.12.026

Levy, B. J., and Anderson, M. C. (2012). Purging of Memories from Conscious Awareness Tracked in the Human Brain. *Journal of Neuroscience*, *32*(47), 16785–16794. https://doi.org/10.1523/JNEUROSCI.2640-12.2012

Liu, P., Hulbert, J. C., Yang, W., Guo, Y., Qiu, J., and Anderson, M. C. (2021). Task compliance predicts suppression-induced forgetting in a large sample. *Scientific Reports*, *11*(1), 20166. https://doi.org/10.1038/s41598-021-99806-8

Marchewka, A., Żurawski, Ł., Jednoróg, K., and Grabowska, A. (2014). The Nencki Affective Picture System (NAPS): Introduction to a novel, standardized, wide-range, high-quality, realistic picture database. *Behavior Research Methods*, *46*(2), 596–610. https://doi.org/10.3758/s13428-013-0379-1

Mary, A., Dayan, J., Leone, G., Postel, C., Fraisse, F., Malle, C., Vallée, T., Klein-Peschanski, C., Viader, F., de la Sayette, V., Peschanski, D., Eustache, F., and Gagnepain, P. (2020). Resilience after trauma: The role of memory suppression. *Science*, *367*(6479), eaay8477. https://doi.org/10.1126/science.aay8477

Norman, K. A., Polyn, S. M., Detre, G. J., and Haxby, J. V. (2006). Beyond mind-reading: Multi-voxel pattern analysis of fMRI data. *Trends in Cognitive Sciences*, *10*(9), 424–430. https://doi.org/10.1016/j.tics.2006.07.005

Paz-Alonso, P. M., Bunge, S. A., Anderson, M. C., and Ghetti, S. (2013). Strength of Coupling within a Mnemonic Control Network Differentiates Those Who Can and Cannot Suppress Memory Retrieval. *Journal of Neuroscience*, *33*(11), 5017–5026. https://doi.org/10.1523/JNEUROSCI.3459-12.2013

Richter, F. R., Cooper, R. A., Bays, P. M., and Simons, J. S. (2016). Distinct neural mechanisms underlie the success, precision, and vividness of episodic memory. *eLife*, *5*. https://doi.org/10.7554/*eLife*.18260

Sheldon, S., and Levine, B. (2013). Same as it ever was: Vividness modulates the similarities and differences between the neural networks that support retrieving remote and recent autobiographical memories. *NeuroImage*, *83*, 880–891. https://doi.org/10.1016/j.neuroimage.2013.06.082

Staresina, B. P., Duncan, K. D., and Davachi, L. (2011). Perirhinal and Parahippocampal Cortices Differentially Contribute to Later Recollection of Object- and Scene-Related Event Details. *Journal of Neuroscience*, *31*(24), 8739–8747. https://doi.org/10.1523/JNEUROSCI.4978-10.2011

Todd, R. M., Schmitz, T. W., Susskind, J., and Anderson, A. K. (2013). Shared Neural Substrates of Emotionally Enhanced Perceptual and Mnemonic Vividness. *Frontiers in Behavioral Neuroscience*, *7*. https://doi.org/10.3389/fnbeh.2013.00040

van Schie, K., Geraerts, E., and Anderson, M. C. (2013). Emotional and non-emotional memories are suppressible under direct suppression instructions. Cognition and Emotion, 27(6), 1122–1131. https://doi.org/10.1080/02699931.2013.765387

Wessa, M., Kanske, P., Neumeister, P., Bode, K., Heissler, J., and Schönfelder, S. (2010). EmoPics: Subjektive und psychophysiologische Evaluation neuen Bildmaterials für die klinisch-bio-psychologische Forschung. Zeitschrift Für Klinische Psychologie Und Psychotherapie, 39(Suppl. 1/11), 77.